# Efficient Convex Relaxations for Streaming PCA

**Raman Arora**
Dept. of Computer Science
Johns Hopkins University
Baltimore, MD 21204
arora@cs.jhu.edu

**Teodor V. Marinov**
Dept. of Computer Science
Johns Hopkins University
Baltimore, MD 21204
tmarino2@jhu.edu

## Abstract

We revisit two algorithms, matrix stochastic gradient (MSG) and $\ell_2$-regularized MSG (RMSG), that are instances of stochastic gradient descent (SGD) on a convex relaxation to principal component analysis (PCA). These algorithms have been shown to outperform Oja's algorithm, empirically, in terms of the iteration complexity, and to have runtime comparable with Oja's. However, these findings are not supported by existing theoretical results. While the iteration complexity bound for $\ell_2$-RMSG was recently shown to match that of Oja's algorithm, its theoretical efficiency was left as an open problem. In this work, we give improved bounds on per iteration cost of mini-batched variants of both MSG and $\ell_2$-RMSG and arrive at an algorithm with total computational complexity matching that of Oja's algorithm.

## 1 Introduction

Principal component analysis (PCA) is a fundamental dimensionality reduction tool used by statisticians and machine learning practitioners alike. In this paper, we study PCA in a streaming setting wherein we receive a stream of high dimensional vectors sampled from an unknown distribution. The goal is to project each point to a lower dimensional space such that most of the information in data, as measured by variance, is preserved.

Formally, we are given a stream of data vectors $(x_t)_{t=1}^T \subset \mathbb{R}^d$, such that each point is sampled i.i.d. from a distribution $x_t \sim \mathcal{D}$, with covariance matrix $C = \mathbb{E}_{x \sim \mathcal{D}}[xx^\top] \in \mathbb{R}^{d \times d}$. Assuming the distribution is zero-mean, the problem is to output an orthonormal $U_t \in \mathbb{R}^{d \times k}$, after observing $x_t$, which tries to minimize $\mathbb{E}_{x \sim \mathcal{D}}[\|UU^\top x - x\|_2^2]$ over all possible orthonormal matrices $U \in \mathbb{R}^{d \times k}$. Equivalently, we are interested in solving the following non-convex stochastic optimization problem in a streaming setting:

$$
\begin{aligned}
\underset{U \in \mathbb{R}^{d \times k}}{\text{maximize}} \quad & \text{Tr}\left(U^\top C U\right) \\
\text{subject to} \quad & U^\top U = I_k
\end{aligned}
\tag{1}
$$

There have been two classes of algorithms that have been proposed to solve Problem 1. One is based on the stochastic power method, also known as Oja's algorithm and is essentially stochastic gradient descent (SGD) on Problem 1 (De Sa et al., 2014; Hardt & Price, 2014; Balcan et al., 2016; Jain et al., 2016; Shamir, 2016a,b; Allen-Zhu & Li, 2017; Li et al., 2018); note, however, that Problem 1 is non-convex. The second approach consists of relaxing the constraint set and reformulating PCA as an equivalent but convex optimization problem. This latter formulation was initially studied by Warmuth & Kuzmin (2008) in the non-stochastic (online) setting and later revisited by Arora et al. (2013) in a stochastic setting. Formally, the equivalent convex problem to Problem 1 is given as

follows:

$$\begin{aligned}
&\underset{P\in\mathbb{R}^{d\times d}}{\text{maximize}} && \text{Tr}\,(\text{PC}) \\
&\text{subject to} && \text{Tr}\,(\text{P}) \le k, 0 \preceq \text{P} \preceq \text{I}, \text{P}^{\top} = \text{P}
\end{aligned} \qquad (2)$$

Stochastic gradient descent on Problem 2 yields what is referred to as matrix stochastic gradient, or MSG, in the existing literature (Arora et al., 2013). MSG and its variants, e.g. $\ell_2$-regularized MSG (RMSG) (Mianjy & Arora, 2018), admit suboptimality guarantees through standard analysis of SGD. This convex relaxation, however, comes at a cost. In particular, it is possible that in the worst case the per-iteration computational cost of the MSG algorithm is of order $O(d^3)$. This is clearly not desirable and far from the efficient per iteration cost of $O(dk)$ for Oja's algorithm.

Although the worst-case runtime of MSG is pessimistic, in practice it has been observed that MSG is efficient and compares favourably to Oja's algorithm in terms of total iteration complexity as well as overall runtime (Arora et al., 2012, 2013; Mianjy & Arora, 2018; Grabowska & Kotłowski, 2018). A potential conjecture, stemming from previous work, is that the efficiency of MSG is due to rank control inherent in MSG updates. In this work, we take a significant step towards unraveling this puzzling phenomenon underlying the efficiency of both the matrix stochastic gradient (MSG) of Arora et al. (2013) and $\ell_2$-regularized MSG algorithm of Mianjy & Arora (2018). It turns out that the rank control of the MSG update is directly related to properties of the true covariance matrix C. We show that simple mini-batching on top of MSG and RMSG, which plays the role of variance reduction for the stochastic gradients, ensures a per iteration complexity of at most $\tilde{O}(\frac{dk^3}{(\lambda_k(\text{C})-\lambda_{k+1}(\text{C}))^2})$ for both algorithms. Combining the improved per iteration cost of mini-batched RMSG, with a careful analysis, we show that the total computational complexity for achieving an $\epsilon$-suboptimal solution for Problem 1 is $\tilde{O}\left(\frac{dk^2}{\epsilon(\lambda_k(\text{C})-\lambda_{k+1}(\text{C}))^2}\min\{d(\lambda_k(\text{C})-\lambda_{k+1}(\text{C})),1\}\right)$. This matches the complexity of Oja's algorithm, up to a factor of $k$, for solving Problem 1 when $\lambda_k(\text{C}) - \lambda_{k+1}(\text{C}) \ge \Omega(1/d)$ and improves on the complexity of Oja in the case when $\lambda_k(\text{C}) - \lambda_{k+1}(\text{C}) \le o(1/(kd))$.

While we use the variance reduction for the stochastic gradients in the classical way, guaranteeing improved objective progress in the proof of Theorem 4.3, it also plays a different and somewhat unusual role. In particular, the variance reduction is needed to guarantee that the iterates remain rank-$k$ projection matrices, which is key in showing all of our results.

## 2 Related Work

The convex relaxation of the PCA problem in Equation (2) can be traced back to the work of Warmuth & Kuzmin (2008) who pose the non-convex PCA formulation in the online learning setting as choosing the best $k$ out of $d$ experts. While somewhat obfuscated, the convex relaxation arises naturally by considering prediction with expert advice. Warmuth & Kuzmin (2008) then solve the problem using the Matrix Exponentiated Gradient (MEG) algorithm, a natural extension of the Hedge algorithm (Freund & Schapire, 1997). In the stochastic setting, MEG needs $O(k\log(d)/\epsilon^2)$ iterations to achieve $\epsilon$-suboptimal solution, however its per iteration cost is $O(d^3)$.

The connection between the two formulations was formally presented in Arora et al. (2013) who also proposed the matrix stochastic gradient (MSG) algorithm which is a variant of stochastic gradient descent on Problem 2. The MSG updates are given as follows

$$\begin{aligned}
\text{P}_{t+\frac{1}{2}} &= \text{P}_t + \eta_t \text{C}_t \\
\text{P}_{t+1} &= \Pi(\text{P}_{t+\frac{1}{2}})
\end{aligned}, \qquad (3)$$

where $\text{C}_t = \text{x}_t\text{x}_t^{\top}$ is an unbiased estimator of the gradient (aka C) of the objective in Problem 2 based on a single sample, $\Pi$ is a projection onto the convex set of constraints $\{\text{P} \in \mathbb{R}^{d\times d} : \text{Tr}\,(\text{P}) = k, 0 \preceq \text{P} \preceq \text{I}_d\}$ with respect to Frobenius norm, and $\eta_t$ is the step size. This algorithm, if implemented carefully, has per iteration complexity of the order $O(d\,\text{rank}(\text{P}_t)^2)$ and has iteration complexity $O(k/\epsilon^2)$. In theory, the rank of $\text{P}_t$ can grow as large as $d$, however empirically the authors observed that the rank did not grow much more than $k$. While in an optimistic scenario, this algorithm is better than MEG, it still has roughly the same iteration complexity for $\epsilon$-suboptimality, which in some regimes is worse than $\tilde{O}(k/(\epsilon(\lambda_k(\text{C}) - \lambda_{k+1}(\text{C}))^2))$ of Oja's algorithm.

A partial resolution to this problem was given by Mianjy & Arora (2018), and comes in the form of considering a regularized convex problem. In particular, the authors consider the following $\ell_2$-regularized PCA problem:

$$
\begin{aligned}
\underset{P \in \mathbb{R}^{d \times d}}{\text{maximize}} \quad & \text{Tr}\,(PC) - \frac{\lambda}{2}\|P\|_F^2 \\
\text{subject to} \quad & \text{Tr}\,(P) \le k, 0 \preceq P \preceq I, P^\top = P
\end{aligned}, \tag{4}
$$

where $\lambda$ is the regularization parameter. It is shown that as long as $\lambda$ is less than the eigengap at $k$, i.e., $\lambda < \lambda_k(C) - \lambda_{k+1}(C)$, solving Problem 4 recovers a solution to Problem 1. Furthermore, because the objective is $\lambda$-strongly convex, the iteration complexity of SGD on the above problem, dubbed RMSG, is of the order $O(k/(\lambda^2 \epsilon))$. The RMSG updates are given as follows:

$$
\begin{aligned}
P_{t+\frac{1}{2}} &= (1 - \eta_t \lambda)P_t + \eta_t C_t \\
P_{t+1} &= \Pi(P_{t+\frac{1}{2}}).
\end{aligned} \tag{5}
$$

Even though RMSG matches the iteration complexity of Oja's algorithm, it suffers the same worst case per iteration complexity as MSG. Again, it is demonstrated empirically that the rank of the iterates of RMSG do not grow significantly beyond $k$, making the algorithm efficient in practice.

Thus, a natural question to ask is: "Can an algorithm based on a convex relaxation of PCA be shown to have good per iteration complexity?" Or, do we necessarily have to pay a price in terms of the overall computational cost? A recent work of Garber (2018) addresses this question partly when analyzing the Oja's algorithm for a mixed setting of adversarially and stochastically generated data. In particular, the authors show that a slightly modified version of MSG achieves per-iteration complexity of the order $\tilde{O}(d/(\lambda_k(C) - \lambda_{k+1}(C))^2)$; however, the proposed analysis works only for the case when $k = 1$ and the modifications of the algorithm require a warm start initialization $P_1$, together with variance reduced gradients $C_t$ (Garber, 2018). Our work builds on these ideas and we extend these results to arbitrary $k$ for slight modifications of both MSG and RMSG. We note that, even though the algorithms we study use the same variance reduction and warm start tricks, our proof techniques are different from Garber (2018). In particular, we leverage the recently developed high probability convergence results for the last iterate of SGD (Harvey et al., 2018) to guarantee that each intermediate iterate is a rank-$k$ matrix.

Finally, we would like to note that there has been a vast number of papers solving a somewhat related problems of matrix sketching and low rank approximation in streams, however, to the extent of our knowledge these works differ from ours in two significant ways – they do not assume that data is sampled i.i.d. from a distribution and hence, their guarantees are much weaker than ours. Since the goal of this paper is to solve the problem described in Section 1, we do not discuss such works further.

## 3   Notation

We use bold-face lower-case letters to denote vectors $x \in \mathbb{R}^d$, bold-face upper-case letters to denote matrices $A \in \mathbb{R}^{d \times d}$, $I_d$ denotes the $d \times d$ identity matrix. For matrices $A \in \mathbb{R}^{d \times n_1}$ and $B \in \mathbb{R}^{d \times n_2}$, $[A, B] \in \mathbb{R}^{d \times (n_1 + n_2)}$ denotes the matrix formed by appending the columns of B to the columns of A. We use $\|\cdot\|$ to denote the $\ell_2$ norm of a vector and the 2-norm of a matrix and use $\|\cdot\|_F$ to denote the Frobenius norm of a matrix. $\text{Tr}\,(\cdot)$ denotes the trace operator and $\langle A, B \rangle = \text{Tr}\,(A^\top B)$ denotes the standard inner product between matrices. The convex set of constraints is $\mathcal{P}_k = \{P \in \mathbb{R}^{d \times d} : \text{Tr}\,(P) = k, 0 \preceq P \preceq I_d\}$ and the projection onto the set $\mathcal{P}_k$ with respect to Frobenius norm is denoted as $\Pi(\cdot)$. $A \preceq B$ denotes that A is less than B in the positive-semidefinite order. We use $\lambda_k(A)$ to denote the $k$-th eigenvalue of A and $\Delta(A) = \lambda_k(A) - \lambda_{k+1}(A)$ to denote the eigengap at $k$. Asymptotic notation with a tilde on top, e.g. $\tilde{O}$ or $\tilde{\Omega}$, hides poly-logarithmic factors. The operator Top-k(A) returns a projection matrix onto the span of the eigenvectors of A corresponding to the top $k$ eigenvalues of A.

---

**Algorithm 1** Mini-batched MSG (MB-MSG)

---

**Input:** Stream of data $\{x_{t_l}\}$ of $d$-dimensional vectors, parameters $\Delta(C)$, probability of failure $\delta$, number of components $k$

**Output:** $P_T \in \mathbb{R}^{d \times d}$

1: $n = \tilde{\Omega}\left(\frac{k^2}{\Delta(C)^3}\right)$

2: $P_1 = \text{Top-k}(\frac{1}{n}\sum_{l=1}^{n} x_{0_l}x_{0_l}^\top)$          %% $\{x_{0_l}\}_{l=1}^n$ is the warm-start mini-batch

3: $n = \tilde{\Omega}\left(\frac{k^3}{\Delta(C)^2}\right)$

4: **for** $t = 1, \ldots, T-1$ **do**

5:     $\eta_t = \tilde{O}\left(1\big/\sqrt{t + \frac{k^2}{\Delta(C)^2}}\right)$

6:     $C_t \leftarrow \frac{1}{n}\sum_{l=1}^{n} x_{t_l}x_{t_l}^\top$          %% $\{x_{t_l}\}_{l=1}^n$ is the mini-batch for the $t^{th}$ epoch

7:     $P_{t+\frac{1}{2}} \leftarrow P_t + \eta_t C_t$

8:     $P_{t+1} = \Pi(P_{t+\frac{1}{2}})$

9: **end for**

---

## 4 Algorithm and Main Result

For simplicity of presentation, we assume that $\|x_t\| \leq 1$ for all $t$, and that $\|C - C_t\|_F \leq 1$. The first assumption implies that $\lambda_1(C) \leq 1$ and $\lambda_1(C_t) \leq 1$. These assumptions are somewhat benign, and primarily for notational convenience when stating the main results and writing the proofs; these are also standard in previous analyses of Oja's algorithm. We also note that the algorithms proposed here require the knowledge of the eigengap $\Delta(C)$. While knowing the exact eigengap is unlikely in practical scenarios, we treat the eigengap as a hyperparameter that can be tuned on a grid. We emphasize that even Oja's algorithm requires the knowledge of the eigengap.

### 4.1 Mini-batched MSG (MB-MSG)

We begin with a variant of MSG (pseudocode given in Algorithm 1) with two simple modifications. First, we initialize $P_1$ sufficiently close to the optimal solution $P^*$, and second, we use mini-batches to form a variance reduced estimate of $C_t$ based on multiple samples. We note that the resulting algorithm does not improve over Oja's algorithm; however, it helps illustrate the techniques that form the basis for the design of the main algorithm in the next section (pseudocode in Algorithm 2).

We initialize the proposed algorithms with a warm start, with the iterate $P_1$ set to the projection matrix onto the span of top-$k$ eigenvectors of the empirical covariance matrix, computed using $\tilde{\Omega}\left(\frac{k^2}{\Delta(C)^3}\right)$ samples. The stream is then broken into epochs, each of size $\tilde{\Omega}\left(\frac{k^3}{\Delta(C)^2}\right)$. We compute the estimate of the gradient, $C_t$, based on the minibatch at the $t^{th}$ epoch, and perform an update of MSG. This ensures that $C_t$ is close enough to $C$ so that we can guarantee each of the iterates remain rank $k$. The step size also slightly differs from $\eta_t = \frac{1}{\sqrt{T}}$, used in the vanilla MSG routine. Such a step size is needed because of the warm start initialization, together with guarantees for the final iterate convergence.

We refer to Algorithm 1 as mini-batched MSG (MB-MSG). It enjoys the following guarantee.

**Theorem 4.1.** *The following holds for Algorithm 1: with probability at least* $1 - \delta$, *for all* $t \leq T$

$$\langle P^* - P_t, C \rangle \leq O\left(\frac{k^4 \log(1/\delta)(\log(T))^2}{\sqrt{t + \frac{1}{\gamma}}}\right),$$

*where* $\gamma = O\left(\frac{\Delta(C)^2}{(k\log(1/\delta))^2}\right)$. *Further, it holds that* $P_t$ *is a rank-$k$ projection matrix.*

The above theorem improves over the result in Arora et al. (2013) in three ways. First, it guarantees the convergence of the last iterate whereas the previous results for MSG have only been for the average iterate. Second, it is a high probability bound, while the previous results for MSG have only

---

**Algorithm 2** Mini-batched $\ell_2$-Regularized MSG (MB-RMSG)

---

**Input:** Stream of data $\{\mathrm{x}_{t_l}\}$ of $d$-dimensional vectors, parameters $\Delta(\mathrm{C})$, probability of failure $\delta$, number of components $k$

**Output:** $\mathrm{P}_T \in \mathbb{R}^{d \times d}$

1: $n = \log\left(3ed/\delta\right) \frac{128 k \log(3e/\delta)}{\Delta(\mathrm{C})^5}$

2: $\mathrm{P}_1 = \text{Top-k}\left(\frac{1}{n} \sum_{l=1}^{n} \mathrm{x}_{0_l} \mathrm{x}_{0_l}^\top\right)$          %% $\{\mathrm{x}_{0_l}\}_{l=1}^{n}$ is the warm-start mini-batch

3: $n = \log\left(\frac{T3ed}{\delta}\right) \frac{8(k+1)^2}{\Delta(\mathrm{C})^2}$

4: **for** $t = 1, \ldots, T-1$ **do**

5:     $\eta_t = \dfrac{1}{\frac{\Delta(\mathrm{C})}{2}\left(t + \frac{128 \log\left(\frac{1}{\delta}\right)}{\Delta(\mathrm{C})^3}\right)}$

6:     $\mathrm{C}_t \leftarrow \frac{1}{n} \sum_{l=1}^{n} \mathrm{x}_{t_l} \mathrm{x}_{t_l}^\top$          %% $\{\mathrm{x}_{t_l}\}_{l=1}^{n}$ is the mini-batch for the $t^{th}$ epoch

7:     $\mathrm{P}_{t+\frac{1}{2}} \leftarrow (1 - \frac{\Delta(\mathrm{C})}{2}\eta_t)\mathrm{P}_t + \eta_t \mathrm{C}_t$

8:     $\mathrm{P}_{t+1} = \Pi(\mathrm{P}_{t+\frac{1}{2}})$

9: **end for**

---

been in expectation. Lastly, it guarantees that every iterate $\mathrm{P}_t$ is rank $k$. Compared to MSG, however, MB-MSG has a higher sample complexity, due to mini-batching at every iteration.

### 4.2 Mini-batched RMSG (MB-RMSG)

Next, we propose and study the mini-batched variant of RMSG, which we refer to as MB-RMSG, detailed in Algorithm 2. MB-RMSG follows the same meta-algorithm as MB-MSG except that it builds on the $\ell_2$-regularized MSG rather than MSG. Again, we initialize $\mathrm{P}_1$ sufficiently close to $\mathrm{P}^*$ and then use mini-batches to reduce the variance of $\mathrm{C}_t$. The update on line 7 is an iteration of SGD on the regularized objective in Equation 4, with $\lambda = \Delta(\mathrm{C})/2$. The choice of $\lambda$ ensures that the solutions to Problem 1 and Problem 4 are identical, as stated in Lemma 2.2 of Mianjy & Arora (2018). Our main result is the following high probability bound for MB-RMSG.

**Theorem 4.2.** *The following holds for Algorithm 2: with probability at least $1 - \delta$, for all $t \leq T$*

$$\langle \mathrm{P}^* - \mathrm{P}_t, \mathrm{C}\rangle \leq \frac{32 \log\left(3e/\delta\right)}{\Delta(\mathrm{C})^2 \left(t + \frac{1}{\gamma} - 1\right)},$$

*where $\gamma = \frac{\Delta(\mathrm{C})^3}{128 \log(1/\delta)}$. Further, for all $t \leq T$ it holds that $\mathrm{P}_t$ is a rank-$k$ projection matrix.*

As with Theorem 4.1, the above result improves on those in Mianjy & Arora (2018) by giving both a high-probability bound on the convergence rate and guaranteeing that each iterate $\mathrm{P}_t$ has rank $k$.

**Computational Cost.** A naive implementation of MB-RMSG requires $O(d^2 k^3/\Delta(\mathrm{C})^2)$ operations per epoch. However, a careful implementation of Algorithm 2 where we maintain an up-to-date singular value decomposition (SVD) of rank-$k$ iterates, requires $O(dk^3/\Delta(\mathrm{C})^2)$ operations per epoch. Then, Theorem 4.2 implies that the total computational complexity to achieve $\epsilon$-suboptimality is $\tilde{O}(dk^3/(\epsilon\Delta(\mathrm{C})^4))$, which is a factor of $k^2\Delta(\mathrm{C})^4$ worse than that of Oja's complexity of $\tilde{O}(dk/(\epsilon\Delta(\mathrm{C})^2))$. Using arguments from proximal theory (Allen-Zhu, 2017), together, with the guarantee that $(\mathrm{P}_t)_{t=1}^{T}$ is a sequence of rank-$k$ projection matrices, we can further leverage the variance reduction in gradient updates to give the following bound.

**Theorem 4.3.** *Let $\mathcal{A}$ be the event that for all $t \in [T]$ it holds that $\|\mathrm{C}_t - \mathrm{C}\| \leq \frac{\Delta(C)}{8(k+1)}$ and $\mathrm{P}_t$ is a rank-$k$ projection matrix. Then Algorithm 2 guarantees that $\mathcal{A}$ occurs with probability at least $1 - \delta$ and that*

$$\mathbb{E}\left[\langle \mathrm{P}^* - \mathrm{P}_T, \mathrm{C}\rangle | \mathcal{A}\right] \leq \tilde{O}\left(\frac{\Delta(\mathrm{C})}{T} + \min\{d\,\Delta(\mathrm{C}), 1\}\frac{1}{kT}\right).$$

Our assumptions on the distribution $\mathcal{D}$ imply that $\Delta(\mathrm{C}) \leq 1/k$. In this case, Theorem 4.3 implies that the total computational complexity for achieving $\epsilon$-suboptimality is $\tilde{O}\left(\frac{dk^2}{\epsilon\Delta(\mathrm{C})^2} \min\{d\Delta(\mathrm{C})), 1\}\right)$,

which is only a factor of $k$ away from Oja's algorithm whenever the gap is large, and actually improves by a factor of $1/\Delta(C)$ over Oja's in the case when $\Delta(C) \in o(1/kd)$.

# 5 Proof sketch

The proofs of both Theorem 4.1 and Theorem 4.2 follow the same ideas. In both cases, essentially, we first establish a sufficient condition for $P_{t+1}$ to be rank $k$, given that $P_t$ is rank $k$. The idea behind this condition is based on Lemma 2 in Garber (2018) and is the following. If $P_t$ captures the subspace spanned by the eigenvectors of $C_t$ corresponding to the top $k$ eigenvalues, then the top $k$ eigenvalues of $P_{t+\frac{1}{2}}$ would be much larger than $\lambda_{k+1}(P_{t+\frac{1}{2}})$. This in turn is sufficient for the projection operator $\Pi$ to set $\lambda_{k+1}(P_{t+\frac{1}{2}})$ to 0. Formally, we show the following for MB-MSG.

**Lemma 5.1.** *Suppose $P_t$ is rank $k$. If $\langle P_t, C_t \rangle + \lambda_k(U_t^\top C_t U_t) \geq \sum_{l=1}^{k+1} \lambda_l(C_t)$, then $P_{t+1}$ is also rank $k$.*

Since it is hard to directly prove that the sufficient condition holds for $P_t$ and $C_t$, we translate the condition to a bound on the suboptimality, i.e., $\langle P^* - P_t, C \rangle \leq \alpha \Delta(C)$, for some constant $\alpha$.

**Lemma 5.2.** *Suppose $\|C - C_t\| \leq \beta$ and $P_t$ is rank $k$. If*

$$\langle P^* - P_t, C \rangle \leq \frac{\Delta(C)}{2} - \beta(k+1),$$

*then $P_{t+1}$ is also rank $k$.*

A similar result for MB-RMSG is given in Lemma B.1 in Appendix B. We know that the condition holds for sufficiently large $t$ from the analysis of SGD and SGD for strongly convex functions (Harvey et al., 2018). The task that remains is to show that the suboptimality bound holds for small $t$. We achieve this by showing that if the first iterate of SGD is initialized from a warm start and the step size is rescaled appropriately, then the following iterates will only improve on the warm start initialization. In the case when the objective is not strongly convex, we additionally need the gradients to be variance reduced to control a certain martingale difference for the initial few terms. This does not contribute to the overall cost of the algorithms, because the variance reduction is anyway needed when translating from the sufficient condition on $C_t$ to the suboptimality condition.

We would like to remark that the above approach is different from the one in Garber (2018), where the rank control is due to a recurrence relation between $\langle P_{t+1}, P^* \rangle$ and $\langle P_t, P^* \rangle$. To the best of our knowledge and attempts, this relation is not easily extendable to the general $k$-components case.

# 6 Implementation details

We focus our discussion on implementing Algorithm 2, however, all of our remarks hold for Algorithm 1 as well. A naive implementation of the algorithm is to form $C_t$ and $P_t$ directly. This already requires $O(d^2)$ space and roughly $\tilde{O}(d^2/\Delta(C)^2)$ computation. The projection operation $\Pi$ also requires taking the eigendecompostion of $P_{t+\frac{1}{2}}$ which is at worst done in time $\tilde{O}(dk^4/\Delta(C)^4)$ because the rank of $P_{t+\frac{1}{2}}$ can grow as large as $\tilde{O}(k^2/\Delta(C)^2)$. Even when one applies the trick in (Arora et al., 2013; Mianjy & Arora, 2018) to always maintain the eigendecomposition of $P_t$ and perform a rank-($C_t$) update as in Brand (2006), the cost is still $\tilde{O}(d \times \text{rank}(C_t)^2) = O(dk^4/\Delta(C)^4)$.

To improve our algorithm, we can take advantage of the fact that the projection always returns a rank-$k$ projection matrix. In particular, $\Pi$ works in the following way. Once given $P_{t+\frac{1}{2}}$, it finds indices $i^*$ and $j^*$ such that $\lambda_i(P_{t+1}) = 1$ for all $i \leq i^*$ and $\lambda_j(P_{t+1}) = 0$ for all $j \geq j^*$. After identifying these indices, $\Pi$ computes a shift $s_{i^*,j^*}$ and sets $\lambda_l(P_{t+1}) = \lambda_l(P_{t+\frac{1}{2}} - s_{i^*,j^*})$, for $i^* + 1 \leq l \leq j^* - 1$, such that $\sum_{l=i^*+1}^{j^*-1} \lambda_l(P_{t+1}) = k - i^*$. Once the condition of Lemma B.1 is met, we know that $i^* = k$ and $j^* = k + 1$ and so $\Pi(P_{t+\frac{1}{2}})$ returns the projection onto the space spanned by the eigenvectors corresponding to the top $k$ eigenvalues of $P_{t+\frac{1}{2}}$. Let $P_t = U_t U_t^\top$ and write $C_t = X_t X_t^\top$, where $X_t \in \mathbb{R}^{d \times n}$ with the $i$-th column equal to $\frac{x_{t_i}}{\sqrt{n}}$, and $n$ is the size of the

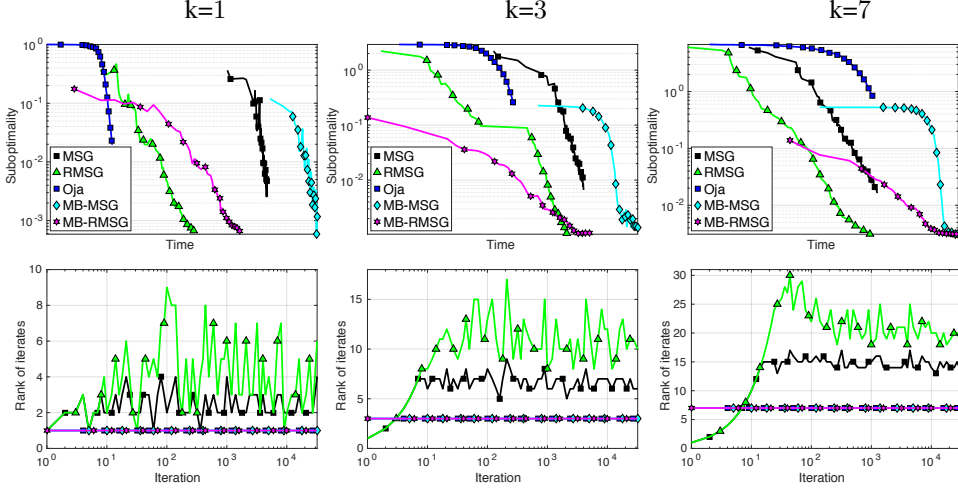

Figure 1: Experiments on synthetic data.

mini-batch. This amounts to changing line 7 in Algorithm 2 to the following:

$$U_{t+1} = \text{Top-k}\left(\left[\sqrt{1 - \Delta(C)\eta_t/2}U_t, \sqrt{\eta_t}X_t\right]\right).$$

This changes the per iteration cost to $\tilde{O}(\frac{dk^3}{\Delta(C)^2})$. Additionally, because we only used the fact that $\|C_t - C\| \leq \gamma$ in the proof of the sufficient condition, we can have an optimistic version of Algorithm 2, where we only need the size of the mini-batch to be large enough, so that the following is satisfied:

$$\langle P_t^* - P_t, C_t \rangle \leq \frac{\Delta(C_t)}{4}, \tag{6}$$

where $P_t^*$ is the projection onto the subspace spanned by the top-$k$ eigenvectors of $C_t$. This follows from the proof of Lemma B.1. The optimistic version is implemented by checking if Equation 6 is satisfied. If it is satisfied, then one proceeds to do the update with the current mini-batch. If it is not satisfied, we double the samples until the condition is satisfied or the mini-batch size becomes greater than the prescribed size on Line 5 of Algorithm 2.

## 7 Empirical results

We include experiments on synthetic data as proof of concept. We also propose more practical variants of MB-MSG and MB-RMSG, which however, do not have theoretical guarantees. Suboptimality is expressed in terms of $\langle P^* - P_t, C \rangle$, where $P^*$ and $C$ are calculated over a test set. We present plots of total runtime to achieve $\epsilon$-suboptimality, and rank of the iterates throughout the iterations. The $x$-axis of the plots is taken to be on a logarithmic scale. We use the k-SVD routine implemented by Liu et al. (2013).

### 7.1 Synthetic data

We generate synthetic data with a large eigengap in the following way. The data is sampled from a multi-variate Normal distribution with zero mean and diagonal covariance matrix $\Sigma$. For each value of $k$, we have $\Sigma_{i,i} = 1$ for $1 \leq i \leq k$ and $\Sigma_{i,i} = gap \times 2^{-i \times 0.1}$ for $k + 1 \leq i \leq d$. In our experiments $gap = 0.1$, $k \in \{1, 3, 7\}$ and $d = 1000$.

The empirical results on the synthetic data set can be found in Figure 1. We use the efficient implementation of MB-MSG and MB-RMSG discussed in Section 6. We also use the sufficient condition stated in Lemma 5.1 for MB-MSG and a similar sufficient condition for MB-RMSG. This allows us to generate mini-batches for $C_t$, with size which is less than the worst case possible, as specified in Algorithm 1 and Algorithm 2. The average mini-batch size for the respective number of components, resulting from the experiments, is given in Table 1.

|       | MB-MSG | MB-RMSG |
|-------|--------|---------|
| k=1   | 7.62   | 6.69    |
| k=3   | 26.72  | 25.30   |
| k=7   | 81.91  | 62.66   |

Table 1: Average mini-batch size on synthetic data.

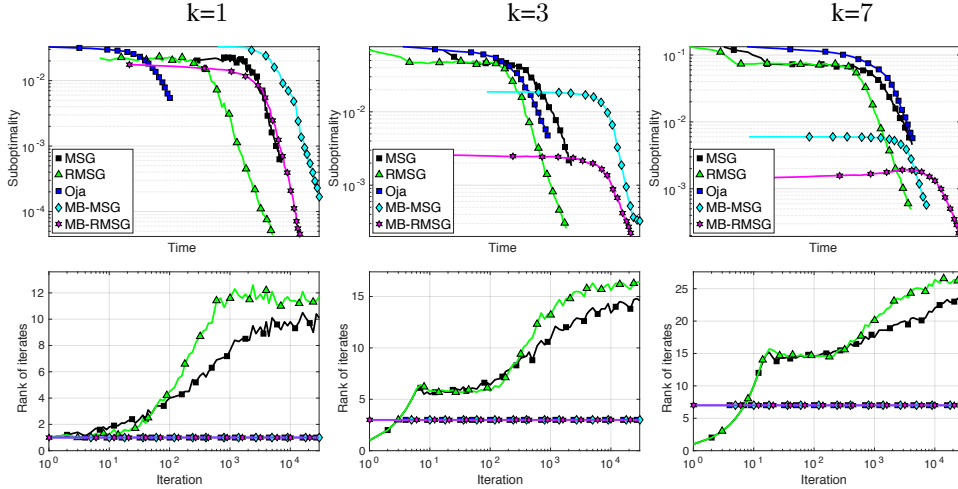

Figure 2: Experiments on MNIST.

We note that we did *not* tune the initial step size for any of the algorithms but rather set step size as recommended by theory. This is because the aim of the experiments is to show that MB-MSG and MB-RMSG satisfy the conditions of Theorem 4.1 and Theorem 4.2.

We see that the average rank of the MSG and RMSG iterates is lower than the average mini-batch size of MB-MSG and MB-RMSG found in Table 1, which determines the per iteration cost of the mini-batched algorithms. This suggests that the total computational complexity of MSG and RMSG is lower than MB-MSG and MB-RMSG. Overall the mini-batched versions of MSG and RMSG stay competitive with their counterparts.

## 7.2 MNIST

We now present empirical results on the MNIST dataset (LeCun, 1998) for a more practical variant of algorithms 1 and 2. The plots can be found in Figure 2. The experiments are carried out for $k \in \{1, 3, 7\}$. The dataset has $d = 784$ and the eigengap between $k$ and $k + 1$ is decreasing exponentially quickly. Instead of setting the maximal mini-batch size in accordance with the theory, we set it to only $1\%$ of the suggested mini-batch size. This violates the sufficient conditions and in practice leads to $\mathrm{rank}(\Pi(\mathrm{P}_{t+\frac{1}{2}})) > k$. However, due to the nature of the efficient version of the algorithms, the rank of $\mathrm{P}_t$ can never grow above $k$. Figure 2 shows that the runtime of MB-MSG and MB-RMSG remains comparable to the runtime of MSG and RMSG.

## 8 Discussion

We present two algorithms based on a convex relaxation to the PCA problem, with convergence guarantees for both of them, which improve on previously known results. We further show that the better of the two algorithms, Algorithm 2, almost matches the total computational complexity of Oja's algorithm, for reaching an $\epsilon$-suboptimal solution in the regime where $\Delta(\mathrm{C})$ is large, and outperforms Oja's algorithm when $\Delta(\mathrm{C}) \leq o(1/(kd))$. We note that the performance guarantees we give are in terms of objective, while the guarantees for Oja's algorithm have classically been in terms of angle between output subspace and best subspace. We do not exclude the possibility that a different style of analysis for Oja's algorithm would guarantee the improved rates we achieve in the setting when eigengap is small. Algorithmic ideas presented here can be applied to improve overall computational complexity of algorithms based on convex relaxations of related subspace learning methods based on partial least squares (Arora et al., 2016) and canonical correlation analysis (Arora et al., 2017).

**Lower bound in Allen-Zhu & Li (2017).** Theorem 6 in Allen-Zhu & Li (2017) implies that any algorithm which returns an orthonormal $U_T \in \mathbb{R}^{d \times k}$ such $\|U_T^\top (U^*)^\perp\|_F^2 \leq O(\epsilon k/\Delta(C)^2)$, has to see at least $1/\epsilon$ samples. Our bound in Theorem 4.3 implies that we can have $\langle P^* - P_T, C \rangle \leq \tilde{O}(\epsilon k/\Delta(C)^2)$ with only $dk\Delta(C)/\epsilon$ samples. We note that this is not a contradiction even when $\Delta(C) \leq o(1/(dk))$, since our upper bound is in terms of objective and not angle between subspaces.

**Relaxing sufficient conditions to $k' > k$.** Our initial goal was to analyze the rank behavior of MSG and RMSG. However, we only managed to analyze a modified version of these algorithms. A first step in proceeding forward is to come up with versions of Lemmas 5.2 and B.1 where we allow the rank of $P_t$ to grow to $k' > k$. Unfortunately our proof techniques do not yield meaningful bounds in this case, as the structure of $P_{t+\frac{1}{2}}$ does not retain some vital properties, whenever $P_t$ is not a projection matrix. We leave developing such sufficient conditions as future work.

## Acknowledgements

This research was supported, in part, by NSF BIGDATA grants IIS-1546482 and IIS-1838139.

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
