[Supplementary Material]

# Supplementary Material to "Efficient Convex Relaxations for Streaming PCA"

We note that constants $\gamma$ and $\beta$ are replaced in the appendix by $\epsilon$ or $\tilde{\epsilon}$. These $\epsilon$'s are not to be confused by the $\epsilon$ used to denote suboptimality, but rather denote a small positive value. We also note that we use $\delta(A)$ to denote the eigengap of $A$ instead of $\Delta(A)$.

## A Proofs for MB-MSG

### A.1 Sufficient condition

**Lemma A.1.** *Suppose $P_t$ is rank $k$. If $\langle P_t, C_t \rangle + \lambda_k(U_t^\top C_t U_t) \geq \sum_{l=1}^{k+1} \lambda_l(C_t)$, then $P_{t+1}$ is also rank $k$.*

*Proof of Lemma 5.1.* Let $P_{t+1/2} = U_{t+1/2} \begin{pmatrix} 1+\lambda_1 & & & & & & \\ & 1+\lambda_2 & & & & & \\ & & \ddots & & & & \\ & & & \lambda_{k+1} & & & \\ & & & & \ddots & \\ & & & & & \lambda_d \end{pmatrix} U_{t+1/2}^\top.$

Let $y = \begin{pmatrix} 1+\lambda_1 \\ 1+\lambda_2 \\ \vdots \\ \lambda_d \end{pmatrix}$ and let $i^*$ and $j^*$ be respectively the largest index of $y$ which is set to 1 after

shifting and the smallest index of $y$ which is set to 0 after shifting. Let the shifting amount for indices $i^*$ and $j^*$ be $s_{i^*,j^*}$. WLOG assume that $\lambda_1 \geq \lambda_2 \geq \ldots \geq \lambda_k$ and $\lambda_{k+1} \geq \ldots \geq \lambda_d$. The projection formula tells us that

$$\sum_{l=i^*+1}^{j^*-1} (y_l - s_{i^*,j^*}) + i^* = k.$$

Thus $s_{i^*,j^*} = \frac{\sum_{l=i^*+1}^{j^*-1} y_l - k + i^*}{j^* - i^* - 2}$. Since we are trying to derive a sufficient condition for when $P_{t+1}$ is rank $k$, we necessarily have $j^* = k + 1$ and thus $s_{i^*,j^*} = \frac{\sum_{l=i^*+1}^{j^*-1} y_l - k + i^*}{k - i^* - 1}$. By the projection rule we must have $s_{i^*,j^*} \geq y_{k+1}$ and thus a sufficient condition becomes

$$\frac{\sum_{l=i^*+1}^{j^*-1} y_l - k + i^*}{k - i^* - 1} = \frac{\sum_{l=i^*+1}^{k} y_l - k + i^*}{k - i^* - 1} = \frac{\sum_{l=i^*+1}^{k} \lambda_l}{k - i^* - 1} \geq \frac{\sum_{l=i^*+1}^{k} \lambda_l}{k - i^*} \geq \lambda_{k+1}. \quad (7)$$

From our assumption on the ordering on $\lambda_l's$ we see that the above sufficient condition is always implied by $\lambda_k \geq \lambda_{k+1}$, since $\frac{\sum_{l=i^*+1}^{k} \lambda_l}{k - i^*} \geq \lambda_k$. This is equivalent to $\lambda_k(P_{t+1/2}) \geq 1 + \lambda_{k+1}(P_{t+1/2})$. We now write the $k$-th eigenvalue of $P_{t+1/2}$ in its variational form $\lambda_k(P_{t+1/2}) = \max_{U \in SO(k)} \min_{u \in \text{span}(U)} u^\top P_{t+1/2} u = \max_{U \in SO(k)} \lambda_{min}(U^\top P_{t+1/2} U) \geq \lambda_k(U_t^\top P_{t+1/2} U_t) = 1 + \eta_t \lambda_k(U_t^\top C_t U_t)$. We also have $\sum_{l=1}^{k} \lambda_l(P_{t+1/2}) = \max_{U \in SO(k)} \text{Tr}\left(U^\top P_{t+1/2} U\right) \geq \text{Tr}\left(U_t^\top P_{t+1/2} U_t\right) = k + \eta_t \sum_{l=1}^{k} \lambda_l(U_t^\top C_t U_t)$. We now use the above inequality to derive:

$$\begin{aligned}
\lambda_{k+1}(P_{t+1/2}) &\leq \sum_{l=1}^{k+1} \lambda_l(P_{t+1/2}) - k - \eta_t \sum_{l=1}^{k} \lambda_l(U_t^\top C_t U_t) \\
&\leq k + \eta_t \sum_{l=1}^{k+1} \lambda_l(C_t) - k - \eta_t \sum_{l=1}^{k} \lambda_l(U_t^\top C_t U_t),
\end{aligned} \quad (8)$$

where the second inequality follows from Ky Fan's inequality and the fact that $P_t$ is a rank $k$ projection matrix so that $\sum_{l=1}^{k+1}(P_t) = k$. The above inequality implies that a sufficient condition for projecting back to a rank-$k$ matrix is $1 + \eta_t \sum_{l=1}^{k+1} \lambda_l(C_t) - \eta_t \sum_{l=1}^{k+1} \lambda_l(U_t^\top C_t U_t) \leq \lambda_k(P_t)$. Together with $\lambda_k(P_{t+1/2}) \geq 1 + \eta_t \lambda_k(U_t^\top C_t U_t)$, we get the following sufficient condition:

$$\lambda_k(U_t^\top C_t U_t) \geq \sum_{l=1}^{k+1} \lambda_l(C_t) - \sum_{l=1}^{k} \lambda_l(U_t^\top C_t U_t). \tag{9}$$

$\square$

**Lemma A.2.** *Suppose that $\|C - C_t\| \leq \epsilon$ and that $P_t$ is rank k. If*

$$\langle P^* - P_t, C \rangle \leq \frac{\delta(C)}{2} - \epsilon(k+1),$$

*then $P_{t+1}$ is also rank k.*

*Proof.* We have:

$$\left| \sum_{l=1}^{k} \lambda_l(U_t^\top C_t U_t) + \lambda_k(U_t^\top C_t U_t) - \sum_{l=1}^{k} \lambda_l(U_t^\top C U_t) - \lambda_k(U_t^\top C U_t) \right|$$

$$\leq |\langle P_t, C_t - C \rangle| + \left| \lambda_{k+1}(U_t^\top C_t U_t) - \lambda_{k+1}(U_t^\top C U_t) \right| \leq k \|C - C_t\| + \left| \lambda_{k+1}(U_t^\top C_t U_t) - \lambda_{k+1}(U_t^\top C U_t) \right|$$

$$\leq \epsilon k + \|U_t^\top (C_t - C) U_t\| \leq (k+1)\epsilon.$$

Using the above and Lemma 5.1, we have that a sufficient condition is $\sum_{l=1}^{k} \lambda_l(U_t^\top C U_t) + \lambda_k(U_t^\top C U_t) - \epsilon(k+1) \geq \sum_{l=1}^{k+1} \lambda_l(C_t)$. Using $|\lambda_l(A + B) - \lambda_l(B)| \leq \|B\|$ we also have $|\lambda_l(C) - \lambda_l(C_t)| \leq \|C - C_t\|$ and thus $\sum_{l=1}^{k+1} |\lambda_l(C_t) - \lambda_l(C)| \leq (k+1)\epsilon$. Thus the sufficient condition becomes:

$$\sum_{l=1}^{k} \lambda_l(U_t^\top C U_t) + \lambda_k(U_t^\top C U_t) - 2\epsilon(k+1) \geq \sum_{l=1}^{k+1} \lambda_l(C). \tag{10}$$

We further simplify the above as follows:

$$\langle P_t, C \rangle \geq \langle \bar{P}, C \rangle + \epsilon(k+1)$$
$$\iff \tag{11}$$
$$\langle P_t, C \rangle \geq \langle P^*, C \rangle - \frac{\delta(C)}{2} + \epsilon(k+1)$$

where $\bar{P} = \frac{(U)_{1:k+1}(U)_{1:k+1}^\top + (U)_{1:k-1}(U)_{1:k-1}^\top}{2}$ and $\delta(C)$ is the eigengap of C at $k$. $\square$

## A.2 Convergence of SGD from "warm start"

We now show it is possible to achieve better guarantees for the $t$-th iterate of SGD, with high probability, if we assume $\|x_1 - x^*\|^2 \leq \epsilon$. As a result, we obtain Theorem 4.1.

**Lemma A.3.** *Consider running SGD on the convex function $f$, with iterates $x_t$. Assume that $\|x_1 - x^*\|^2 \leq \sqrt{\epsilon} \leq 1$ and set $\eta_t = \frac{1}{\sqrt{\frac{1}{\epsilon} + t}}$. Assume that for all $t$ we have that the stochastic gradient $g_t$ is bounded as $\|g_t\| \leq V$ and that for all iterates we have $\|x_t - x^*\| \leq R$. Let $\hat{x} = \sum_{t=1}^{T'} x_t$. Then with probability at least $1 - \delta^{\frac{1}{4c}}$ it holds that*

$$f(\hat{x}) - f(x^*) \leq \frac{1}{T'} + \frac{R^2 + V^2 + \log(1/\delta)}{\sqrt{T' + \frac{1}{\epsilon}}},$$

*for $c = 8V^2 R^2$, as long as $T'\epsilon > 4$.*

*Proof.* Let $g_t$ denote the stochastic gradient at time $t$, with $\|g_t\| \leq V$. Begin as in the proof for SGD:

$$\|x_{t+1} - x^*\|^2 \leq \|x_t - x^*\|^2 - 2\eta_t \langle g_t, x_t - x^* \rangle + \eta_t^2 \|g_t\|^2 \leq \|x_t - x^*\|^2 - 2\eta_t \langle g_t, x_t - x^* \rangle + \eta_t^2 V^2$$
$$= \|x_t - x^*\|^2 - 2\eta_t \langle \nabla f(x_t), x_t - x^* \rangle + 2\eta_t \langle \nabla f(x_t) - g_t, x_t - x^* \rangle + \eta_t^2 V^2.$$

Let $\delta_t = \langle \nabla f(x_t) - g_t, x_t - x^* \rangle$. The above implies that

$$f(x_t) - f(x^*) \leq \frac{\|x_t - x^*\|^2 - \|x_{t+1} - x^*\|^2}{2\eta_t} + \delta_t + \frac{\eta_t}{2} V^2.$$

Let $\widehat{x} = \frac{1}{T'} \sum_{t=1}^{T'} x_t$ and suppose that $\|x_t - x^*\|^2 \leq R^2$. We have

$$T'(f(\widehat{x}) - f(x^*)) \leq \frac{\|x_1 - x^*\|^2}{2\eta_1} + R^2 \sum_{t=2}^{T'} \left( \frac{1}{\eta_t} - \frac{1}{\eta_{t-1}} \right) + \sum_{t=1}^{T'} \delta_t + \frac{V^2}{2} \sum_{t=1}^{T'} \eta_t$$

$$\leq \frac{\sqrt{\epsilon}\sqrt{\frac{1}{\epsilon} + 1}}{2} + R^2 \left( \sqrt{\frac{1}{\epsilon} + T'} - \sqrt{\frac{1}{\epsilon}} \right) + \frac{V^2}{2} \int_{\frac{1}{\epsilon}}^{T'} \frac{1}{\sqrt{t}} dt + \sum_{t=1}^{T'} \delta_t$$

$$\leq 1 + R^2 \left( \sqrt{\frac{1}{\epsilon} + T'} - \sqrt{\frac{1}{\epsilon}} \right) + V^2 \left( \sqrt{T'} - \sqrt{\frac{1}{\epsilon}} \right) + \sum_{t=1}^{T'} \delta_t.$$

Let $T = T' + \frac{1}{\epsilon}$. We have $\frac{\sqrt{T} - \sqrt{\frac{1}{\epsilon}}}{T - \frac{1}{\epsilon}} = \frac{1}{\sqrt{T} + \sqrt{\frac{1}{\epsilon}}} < \frac{1}{\sqrt{T}}$. To finish the bound we need to bound with high probability the sum $\sum_{t=1}^{T'} \delta_t$. It is standard to show that $\delta_t$ is a martingale difference and we also have $|\delta_t| \leq 2VR$. We can apply Azuma's inequality to get

$$\mathbb{P} \left[ \sum_{t=1}^{T'} \delta_t \geq \Delta \right] \leq e^{-\frac{\Delta^2}{8T'V^2R^2}}.$$

Let $c = 8V^2 R^2$. We set $\Delta = \left( \sqrt{T'} - \sqrt{\frac{1}{\epsilon}} \right) \sqrt{\log(1/\delta)}$. Then

$$e^{-\frac{\Delta^2}{8T'V^2R^2}} = \delta^{\frac{1}{c}} e^{\log(1/\delta)\frac{1}{c}(2\sqrt{\frac{1}{T'\epsilon}} - \frac{1}{T'\epsilon})}.$$

The functions $2\frac{1}{\sqrt{x}} - \frac{1}{x}$ is decreasing for $x \geq 1$ and at $x = 4$ it takes value $3/4$ and thus $e^{-\frac{\Delta^2}{8T'V^2R^2}} \leq \delta^{\frac{1}{4c}}$. This implies that, with probability $1 - \delta^{\frac{1}{4c}}$, we have $f(\widehat{x}) - f(x^*) \leq 1/T' + \frac{R^2 + V^2 + \log(1/\delta)}{\sqrt{T' + \frac{1}{\epsilon}}}$, and so we have improved the iteration complexity of SGD by $\frac{1}{\epsilon}$ as long as $T'\epsilon \geq 4$. $\square$

If the the stochastic gradients are sufficiently variance reduced, we can show that the above result holds for all $T$. In particular, we have the following lemma for the iterates of MSG.

**Lemma A.4.** *Suppose that* $\|C - C_t\| \leq \sqrt{\frac{\epsilon}{2}}$ *for all t. Let* $\|C_t\|_F \leq V$ *and* $c = 8k^2$. *With probability at least* $1 - \delta^{1/c}$ *it holds that*

$$\langle P^* - \widehat{P}, C \rangle \leq \frac{1}{T} + \frac{4k^2 + V^2 + \log(1/\delta)}{\sqrt{T + \frac{1}{\epsilon}}},$$

*for all* $T \geq 1$*, where* $\widehat{P} = \frac{1}{T} \sum_{t=1}^{T} P_t$

*Proof.* We would like to refine the bound in Lemma A.3 by using the structure of the MSG update. In particular we have that $|\delta_t| = |\langle C - C_t, P_t - P^* \rangle| \leq \|C - C_t\| 2k$. Suppose that $\|C - C_t\| \leq \tilde{\epsilon}$. Let $c = 8k^2$, then an application of Azuma's inequality with $\Delta = \frac{T}{\sqrt{T + \frac{1}{\epsilon}}} \sqrt{\log(1/\delta)}$ yields

$$\mathbb{P} \left[ \sum_{t=1}^{T} \delta_t \geq \frac{T}{\sqrt{T + \frac{1}{\epsilon}}} \sqrt{\log(1/\delta)} \right] \leq \exp \left( -\log(1/\delta) \frac{T}{(T + \frac{1}{\epsilon})\tilde{\epsilon}^2 c} \right).$$

Notice that we are free to choose $\tilde{\epsilon}$ in our algorithm and since $\frac{x}{x+\frac{1}{\epsilon}}$ is an increasing function of $x$, setting $\tilde{\epsilon} = \sqrt{\frac{\epsilon}{2}} \leq \sqrt{\frac{1}{1+1/\epsilon}}$ guarantees that for any $T \geq 1$ with probability $1 - \delta^{1/c}$ the result of Lemma A.3 holds. $\qquad \square$

For our final result, we are going to need $\epsilon = \Theta(gap(\mathrm{C})^2)$ and thus $\tilde{\epsilon} = \Theta(gap(\mathrm{C}))$, which is anyway required by the sufficient condition.

Next we attempt to improve the iteration complexity for the final iterate analysis with high probability. We note that in expectation this is somewhat easy to achieve.

**Lemma A.5.** *With the same assumptions as in Lemma A.3, with probability $1 - \delta^{\frac{1}{4c}}$ it holds that*

$$f(\mathrm{x}_T) - f(\mathrm{x}^*) \leq \frac{1}{T} + \frac{R^2 + V^2 + \log(1/\delta)}{\sqrt{T + \frac{1}{\epsilon}}} + \frac{(R^2 + V^2)(1 + \log(T))}{\sqrt{T + \frac{1}{\epsilon}}}$$

$$+ \sum_{k=1}^{T-1} \frac{1}{k(k+1)} \sum_{t=T-k}^{T} \langle \nabla f(\mathrm{x}_t) - \mathrm{g}_t, \mathrm{x}_t - \mathrm{x}_{T-k} \rangle$$

*Proof.* The proof essentially follows the proof of Theorem 2 in Shamir & Zhang (2013). Let $\delta_{t,k} = \langle \nabla f(\mathrm{x}_t) - \mathrm{g}_t, \mathrm{x}_t - \mathrm{x}_{T-k} \rangle$. We have

$$f(\mathrm{x}_t) - f(\mathrm{x}_{T-k}) \leq \frac{\|\mathrm{x}_t - \mathrm{x}_{T-k}\|^2 - \|\mathrm{x}_{t+1} - \mathrm{x}_{T-k}\|^2}{2\eta_t} + \frac{\eta_t}{2}\|\mathrm{g}_t\|^2 + \delta_{t,k}.$$

The above implies

$$\sum_{t=T-k}^{T} (f(\mathrm{x}_t) - f(\mathrm{x}_{T-k})) \leq \sum_{t=T-k+1}^{T} \frac{1}{2}\|\mathrm{x}_t - \mathrm{x}_{T-k}\|^2 \left( \frac{1}{\eta_t} - \frac{1}{\eta_{t+1}} \right) + \frac{V^2}{2} \sum_{t=T-k}^{T} \eta_t + \delta_{t,k}$$

$$\leq \frac{R^2}{2}\left( \sqrt{T + \frac{1}{\epsilon}} - \sqrt{T - k + \frac{1}{\epsilon}} \right) + \frac{V^2}{2} \int_{t=T-k-1+\frac{1}{\epsilon}}^{T+\frac{1}{\epsilon}} \frac{1}{\sqrt{t}} dt + \sum_{t=T-k}^{T} \delta_{t,k}$$

$$= \frac{R^2}{2}\left( \sqrt{T + \frac{1}{\epsilon}} - \sqrt{T - k + \frac{1}{\epsilon}} \right) + V^2\left( \sqrt{T + \frac{1}{\epsilon}} - \sqrt{T - k - 1 + \frac{1}{\epsilon}} \right) + \sum_{t=T-k}^{T} \delta_{t,k}$$

$$\leq \left( \frac{R^2}{2} + V^2 \right) \frac{k+1}{\sqrt{T + \frac{1}{\epsilon}}} + \sum_{t=T-k}^{T} \delta_{t,k}.$$

Let $S_k = \frac{1}{k+1} \sum_{t=T-k}^{T} f(\mathrm{x}_t)$. From the above we have

$$-f(\mathrm{x}_{T-k}) \leq -S_k + \left( \frac{R^2}{2} + V^2 \right) \frac{1}{\sqrt{T + \frac{1}{\epsilon}}} + \frac{1}{k+1} \sum_{t=T-k}^{T} \delta_{t,k}.$$

By the definition of $S_k$, it also holds that

$$kS_{k-1} = (k+1)S_k - f(\mathrm{x}_{T-k}) \leq (k+1)S_k - S_k + \left( \frac{R^2}{2} + V^2 \right) \frac{1}{\sqrt{T + \frac{1}{\epsilon}}} + \frac{1}{k+1} \sum_{t=T-k}^{T} \delta_{t,k}.$$

The above implies

$$S_{k-1} \leq S_k + \left( \frac{R^2}{2} + V^2 \right) \frac{1}{k\sqrt{T + \frac{1}{\epsilon}}} + \frac{1}{k(k+1)} \sum_{t=T-k}^{T} \delta_{t,k},$$

and thus we have

$$f(\mathrm{x}_T) \leq S_{T-1} + \left( \frac{R^2}{2} + V^2 \right) \frac{1 + \log(T)}{\sqrt{T + \frac{1}{\epsilon}}} + \sum_{k=1}^{T-1} \frac{1}{k(k+1)} \sum_{t=T-k}^{T} \delta_{t,k}.$$

Remembering that $S_{T-1} = \frac{1}{T}\sum_{t=1}^{T} f(\mathbf{x}_t)$, subtracting $f(\mathbf{x}^*)$ from both sides of the inequality and using Lemma A.3 we arrive at the desired result. $\qquad\square$

The rest of our efforts are now focused on bounding the term $\sum_{k=1}^{T-1}\frac{1}{k(k+1)}\sum_{t=T-k}^{T}\langle\nabla f(\mathbf{x}_t) - \mathbf{g}_t, \mathbf{x}_t - \mathbf{x}_{T-k}\rangle$. We now attempt to do so by repeating the proofs in Harvey et al. (2018). We begin by showing that Lemma 8.4 from Harvey et al. (2018) still holds.

**Lemma A.6.** *Assume $\lambda_1(\mathbf{C}) \leq 1$, then*

$$\|\mathbf{x}_a - \mathbf{x}_b\|^2 \leq \sum_{i=a}^{b-1}\frac{\|\mathbf{g}_i\|^2}{i} + 2\sum_{i=a}^{b-1}\frac{f(\mathbf{x}_i) - f(\mathbf{x}_a)}{\sqrt{i}} + 2\sum_{i=a}^{b-1}\frac{\langle\nabla f(\mathbf{x}_i) - \mathbf{g}_i, \mathbf{x}_i - \mathbf{x}_a\rangle}{\sqrt{i}}$$

*Proof.*

$$\begin{aligned}
\|\mathbf{x}_a - \mathbf{x}_b\|^2 &\leq \|\mathbf{x}_a - \mathbf{x}_{b-1} + \eta_{b-1}\mathbf{g}_{b-1}\|^2 = \|\mathbf{x}_a - \mathbf{x}_{b-1}\|^2 + 2\eta_{b-1}\langle\mathbf{g}_{b-1}, \mathbf{x}_a - \mathbf{x}_{b-1}\rangle + \eta_{b-1}^2\|\mathbf{g}_{b-1}\|^2 \\
&\leq \|\mathbf{x}_a - \mathbf{x}_{b-1}\|^2 + 2\eta_{b-1}(f(\mathbf{x}_{b-1}) - f(\mathbf{x}_a)) + 2\eta_{b-1}\langle\nabla f(\mathbf{x}_{b-1}) - \mathbf{g}_{b-1}, \mathbf{x}_{b-1} - \mathbf{x}_a\rangle + \eta_{b-1}^2\|\mathbf{g}_{b-1}\|^2 \\
&\leq \cdots \\
&\leq \sum_{i=a}^{b-1}\eta_i^2\|\mathbf{g}_i\|^2 + 2\sum_{i=a}^{b-1}\eta_i(f(\mathbf{x}_i) - f(\mathbf{x}_a)) + 2\sum_{i=a}^{b-1}\eta_i\langle\nabla f(\mathbf{x}_i) - \mathbf{g}_i, \mathbf{x}_i - \mathbf{x}_a\rangle \\
&< \sum_{i=a}^{b-1}\frac{\|\mathbf{g}_i\|^2}{i} + 2\sum_{i=a}^{b-1}\frac{f(\mathbf{x}_i) - f(\mathbf{x}_a)}{\sqrt{i}} + 2\sum_{i=a}^{b-1}\frac{\langle\nabla f(\mathbf{x}_i) - \mathbf{g}_i, \mathbf{x}_i - \mathbf{x}_a\rangle}{\sqrt{i}},
\end{aligned}$$

where the last inequality follows from the fact that $\eta_i = \sqrt{\frac{1}{i+\frac{1}{\epsilon}}} < \frac{1}{\sqrt{i}}$. $\qquad\square$

We next introduce some additional notation. Let $\alpha_j = \frac{1}{(T-j+1)(T-j)}$, $\mathbf{w}_t = \sum_{j=1}^{t-1}\alpha_j(\mathbf{x}_t - \mathbf{x}_j)$. Change of summation shows that

$$\sum_{k=1}^{T/2}\frac{1}{k(k+1)}\sum_{t=T-k}^{T}\langle\nabla f(\mathbf{x}_t) - \mathbf{g}_t, \mathbf{x}_t - \mathbf{x}_{T-k}\rangle = \sum_{t=T/2}^{T-1}\langle\nabla f(\mathbf{x}_t) - \mathbf{g}_t, \mathbf{w}_t\rangle$$

Now the proof of Lemma 8.6 Harvey et al. (2018) is unchanged. We restate the lemma here of convenience.

**Lemma A.7.** *Assume $f$ is 1-Lipschitz, then there exists a constant $c$ (depending on $V$ and $R$) such that*

$$\sum_{t=T/2}^{T}\|\mathbf{w}_t\|^2 \leq c\frac{(\log(T))^2\sqrt{\log(1/\delta)}}{T} + \sum_{t=T/2}^{T}\langle\nabla f(\mathbf{x}_t) - \mathbf{g}_t, \frac{4\log(T)}{\sqrt{t}}\mathbf{w}_t\rangle,$$

*with probability at least $1 - \delta$.*

We are going to use Lemma A.7 in combination with Corollary C.5 in Harvey et al. (2018), which states the following:

**Lemma A.8.** *Let $\{\mathcal{F}_t\}_t$ be a filtration, let $\mathbf{a}_t$ be an $\mathcal{F}_t$-measurable random vector and let $\mathbf{b}_t$ be an $\mathcal{F}_{t-1}$-measurable random vector. Let $d_t = \langle\mathbf{a}_t, \mathbf{b}_t\rangle$ and suppose $(\mathbf{a}_t)_t$ is a Martingale difference, with $\|\mathbf{a}_t\| \leq 1$. Further suppose that there exist $R > 0$ and a positive sequence $(\alpha_t)_t$ such that $\max_t\{\alpha_t\} \leq c\sqrt{R}$ (for some constant $c$), such that exactly one of the following holds for all $\delta \in (0, 1)$.*

1. *$\sum_{t=1}^{T}\|\mathbf{b}_t\|^2 \leq \sum_{t=1}^{T}\alpha_t d_t + R\log(1/\delta)$ with probability at least $1 - \delta$;*

2. *$\sum_{t=1}^{T}\|\mathbf{b}_t\|^2 \leq \sum_{t=1}^{T}\alpha_t d_t + R\sqrt{\log(1/\delta)}$ with probability at least $1 - \delta$.*

*Then*

$$\mathbb{P}\left[\sum_{t=1}^{T}d_t \geq x\right] \leq \delta + \exp\left(-\frac{x^2}{4\max_t\{\alpha_t\}_{t=1}^{T-1}x + 8R\log(1/\delta)}\right).$$

Combining Lemma A.6 and Lemma A.8 we arrive at the following:

**Lemma A.9.** *Suppose* $\|\nabla f(\mathrm{x}_t) - \mathrm{g}_t\| \leq 1$ *and* $T\epsilon \geq 1$. *Then with probability* $1 - 2\delta^{\frac{1}{2c}}$ *it holds that*

$$\sum_{t=T/2}^{T} \langle \nabla f(\mathrm{x}_t) - \mathrm{g}_t, \mathrm{w}_t \rangle \leq \frac{\log(T)\log(1/\delta)}{\sqrt{T + \frac{1}{\epsilon}}}.$$

*Proof.* Let $\mathrm{a}_t = \nabla f(\mathrm{x}_t) - \mathrm{g}_t$, $\mathrm{b}_t = \mathrm{w}_t$, $\alpha_t = \frac{4\log(T)}{\sqrt{t}}$ for $t = T/2, \ldots, T-1$ and $\alpha_T = 0$. Then $\max_t\{\alpha_t\} \leq c_1\frac{\log(T)}{\sqrt{T}}$. Let $R = c_2\frac{(\log(T))^2}{T}$. Then from Lemma A.8 we have

$$\mathbb{P}\left[\sum_{t=T/2}^{T} \langle \nabla f(\mathrm{x}_t) - \mathrm{g}_t, \mathrm{w}_t \rangle \geq x\right] \leq \delta + \exp\left(-\frac{x^2}{4c_1\frac{\log(T)}{\sqrt{T}}x + 8c_2\frac{(\log(T))^2}{T}\log(1/\delta)}\right).$$

Set $x = \frac{\log(T)}{\sqrt{T+\frac{1}{\epsilon}}}\log(1/\delta)$. Then

$$\exp\left(-\frac{\frac{\log(T)^2}{T+\frac{1}{\epsilon}}\log(1/\delta)^2}{4c_1\frac{\log(T)}{\sqrt{T}}\frac{\log(T)}{\sqrt{T+\frac{1}{\epsilon}}}\log(1/\delta) + 8c_2\frac{(\log(T))^2}{T}\log(1/\delta)}\right) \leq \exp\left(-\log(1/\delta)\frac{T}{c(T+1/\epsilon)}\right),$$

where $c = 12\max(c_1, c_2)$. Thus as long as $T\epsilon \geq 1$ we have with probability $1 - 2\delta^{\frac{1}{2c}}$

$$\sum_{t=T/2}^{T} \langle \nabla f(\mathrm{x}_t) - \mathrm{g}_t, \mathrm{w}_t \rangle \leq \frac{\log(T)\log(1/\delta)}{\sqrt{T + \frac{1}{\epsilon}}}$$

□

**Theorem A.10.** *Suppose that* $\|\mathrm{P}_1 - \mathrm{P}^*\|_F^2 \leq \sqrt{\epsilon}$, *for all t it holds that* $\|\mathrm{C}_t - \mathrm{C}\| \leq \sqrt{\frac{\epsilon}{2}}\frac{1}{\log(1/\epsilon)}$ *and that* $\|\mathrm{C} - \mathrm{C}_t\| \leq 1$. *For any* $T \geq 1$, *with probability* $1 - 4\delta^{1/\tilde{c}}$ *(for some* $\tilde{c} = O(k^2V^2)$*) it holds that*

$$\langle \mathrm{P}^* - \mathrm{P}_T, \mathrm{C} \rangle \leq O\left(\frac{k^2\log(1/\delta)(\log(T))^2}{\sqrt{T + \frac{1}{\epsilon}}}\right).$$

*Proof of Theorem A.10.* Lemma A.4, together with Lemma A.5 gives us that with probability $1 - \delta^{1/c}$

$$\langle \mathrm{P}^* - \mathrm{P}_T, \mathrm{C} \rangle \leq \frac{1}{T} + \frac{4k^2 + V^2 + \log(1/\delta)}{\sqrt{T + \frac{1}{\epsilon}}} + \frac{(4k^2 + V^2)(1 + \log(T))}{\sqrt{T + \frac{1}{\epsilon}}}$$

$$+ \sum_{k=1}^{T-1}\frac{1}{k(k+1)}\sum_{t=T-k}^{T}\langle \mathrm{C} - \mathrm{C}_t, \mathrm{P}_t - \mathrm{P}_{T-k}\rangle.$$

We now bound $\sum_{k=1}^{T-1}\frac{1}{k(k+1)}\sum_{t=T-k}^{T}\langle \mathrm{C} - \mathrm{C}_t, \mathrm{P}_t - \mathrm{P}_{T-k}\rangle$ by splitting it into two parts. First consider $\sum_{k=T/2+1}^{T-1}\frac{1}{k(k+1)}\sum_{t=T-k}^{T}\langle \mathrm{C} - \mathrm{C}_t, \mathrm{P}_t - \mathrm{P}_{T-k}\rangle$. For a fixed $k$, we can use Azuma's inequality with the assumption that $\|\mathrm{C} - \mathrm{C}_t\| \leq \sqrt{\frac{\epsilon}{2}}$ and get as in Lemma A.4 that $\mathbb{P}\left[\sum_{t=T-k}^{T}\langle \mathrm{C} - \mathrm{C}_t, \mathrm{P}_t - \mathrm{P}_{T-k}\rangle \geq \log(T/\delta)\frac{k}{\sqrt{k+\frac{1}{\epsilon}}}\right] \leq \frac{2}{T}\delta^{1/c}$. A union bound for all $k$ thus gives us

$$\mathbb{P}\left[\sum_{k=T/2+1}^{T-1}\frac{1}{k(k+1)}\sum_{t=T-k}^{T}\langle \mathrm{C} - \mathrm{C}_t, \mathrm{P}_t - \mathrm{P}_{T-k}\rangle \geq \log(T/\delta)\sum_{k=T/2+1}^{T-1}\frac{1}{\sqrt{k+1/\epsilon}(k+1)}\right] \leq \delta^{1/c}.$$

Notice that

$$\sum_{k=T/2+1}^{T-1} \frac{1}{\sqrt{k+1/\epsilon}(k+1)} \leq \frac{T}{2}\frac{1}{\sqrt{T/2+\frac{1}{\epsilon}}\frac{T}{2}},$$

and thus

$$\mathbb{P}\left[\sum_{k=T/2+1}^{T-1} \frac{1}{k(k+1)} \sum_{t=T-k}^{T} \langle C - C_t, P_t - P_{T-k}\rangle \geq \frac{2\log(T/\delta)}{\sqrt{T+\frac{1}{\epsilon}}}\right] \leq \delta^{1/c}.$$

Next we bound $\sum_{k=1}^{T/2}\frac{1}{k(k+1)}\sum_{t=T-k}^{T}\langle C - C_t, P_t - P_{T-k}\rangle$ by using Lemma A.9 and a refinement for the case when $T\epsilon < 1$. When $T\epsilon \geq 1$, direct application of Lemma A.9 gives us:

$$\mathbb{P}\left[\sum_{k=1}^{T/2} \frac{1}{k(k+1)} \sum_{t=T-k}^{T} \langle C - C_t, P_t - P_{T-k}\rangle \geq \frac{\log(T)\log(1/\delta)}{\sqrt{T+\frac{1}{\epsilon}}}\right] \leq 2\delta^{1/2c'}.$$

Suppose now that $T\epsilon < 1$. Let $\|C - C_t\| \leq \tilde{\epsilon}$. Then

$$\sum_{k=1}^{T/2} \frac{1}{k(k+1)} \sum_{t=T-k}^{T} \langle C - C_t, P_t - P_{T-k}\rangle < \sum_{k=1}^{1/\epsilon} \frac{\tilde{\epsilon}}{k+1} \leq \log(1/\epsilon)\,\tilde{\epsilon}.$$

Let $\tilde{c} = \max(c, 2c')$. The above implies that as long as $\tilde{\epsilon} \leq \sqrt{\frac{\epsilon}{2}}\frac{1}{\log(1/\epsilon)}$, for any $T \geq 1$

$$\sum_{k=1}^{T-1} \frac{1}{k(k+1)} \sum_{t=T-k}^{T} \langle C - C_t, P_t - P_{T-k}\rangle \leq \frac{\log(T)\log(1/\delta)}{\sqrt{T+\frac{1}{\epsilon}}},$$

with probability at least $1 - 3\delta^{1/\tilde{c}}$. Combining this, with Lemma A.5 and a union bound over $T$ completes the proof. □

## A.3 Putting everything together

Let the $\epsilon$ in Lemma 5.2 be $\epsilon_1$ and the $\epsilon$ in Theorem A.10 be $\epsilon_2$. We are going to set $\epsilon_1 = \frac{\delta(C)}{4(k+1)}$ and $\epsilon_2 = (\frac{\delta(C)}{c})^2$, for an appropriate constant $c$, which depends only on $\|x_{t_n}\|_2, k$ and $\log(1/\delta)$. In particular from Theorem A.10, we know that for some $c_1$ (which depends only on $\|C_t\|_F$ and $k$, and therefore only on $\|x_{t_n}\|$ and $k$) it holds $\langle P^* - P_T, C\rangle \leq c_1\frac{\log(1/\delta)(\log(T))^2}{\sqrt{T+\frac{1}{\epsilon_2}}}$, for all $T$ with probability $1 - 3\delta^{1/c'}$. Setting $\epsilon_1 = \frac{\delta(C)}{4(k+1)}$, Lemma 5.2 implies that we should have $c_1\frac{\log(1/\delta)(\log(T))^2}{\sqrt{T+\frac{1}{\epsilon_2}}} \leq \epsilon_1 = \frac{\delta(C)}{4(k+1)}$. Since $\frac{(\log(T))^2}{\sqrt{T+\frac{1}{\epsilon_2}}}$ is only decreasing in $T$, we have $c_1\frac{\log(1/\delta)(\log(T))^2}{\sqrt{T+\frac{1}{\epsilon_2}}} \leq c_1\sqrt{\epsilon_2}\log(1/\delta)(\log(3))^2$ and thus we can take $\epsilon_2 \leq c_2\delta(C)^2$, where $c_2 = \frac{1}{(4(k+1))^2 c_1^2(\log(1/\delta))^2(\log(3))^4}$. Thus we would like $\|P_1 - P^*\|_F^2 \leq \sqrt{c_2}\delta(C)$ and $\|C_t - C\| \leq \sqrt{c_2}\delta(C)$ for all $t \leq T$. From Davis-Kahan's theorem Yu et al. (2014), we know that $\|P_1 - P^*\|_F^2 \leq \frac{4k\|C-C_1\|^2}{\delta(C)^2}$. Thus we would like $\|C - C_1\|^2 \leq \frac{\sqrt{c_2}\delta(C)^3}{4k}$, together with $\|C - C_t\| \leq \sqrt{c_2}\delta(C)$. We use Matrix Bernstein's Tropp et al. (2015) inequality to satisfy both. In particular, if at every iteration we use $n = O(\frac{\log(T)\log\left(\frac{d}{\delta^{1/c'}}\right)}{c_2\delta(C)^2})$ samples to compute $C_t$ and we use $n = O(\frac{2\sqrt{k}\log\left(\frac{d}{\delta^{1/c'}}\right)}{c_2^{1/4}\sqrt{\delta(C)}gap})$ samples to compute $C_1$, and initialize $P_1$ as the projection on the top-$k$ eigenspace of $C_1$, then after $T$ iterations we have $\langle P^* - P_T, C\rangle \leq O\left(\frac{\log(1/\delta)(\log(T))^2}{\sqrt{T+\frac{1}{\epsilon_2}}}\right)$, with probability $1 - 4\delta^{1/c'}$ and each of the $P_t$'s is a rank $k$ projection matrix.

**Theorem A.11.** *Assume $\|x_{t_n}\|^2 \leq \frac{1}{2}$. After running Algorithm 1, with probability at least $1 - 4\delta^{1/c}$, where $c = O(k^2 V^2)$, it holds that*

$$\langle P^* - P_T, C \rangle \leq O\left( \frac{k^2 \log(1/\delta)(\log(T))^2}{\sqrt{T + \frac{1}{\epsilon}}} \right).$$

*and for all $t \leq T$ it holds that $P_t$ is a rank-$k$ projection matrix, where $\epsilon = O\left( \frac{\delta(C)^2}{(k \log(1/\delta))^2} \right)$.*

*Proof of Theorem 4.1.* We proceed as in the proof of Theorem 4.2. From Lemma 11 we know that as long as $\|C - C_t\| \leq \frac{\delta(C)}{4(k+1)}$, $P_t$ is rank $k$, then $\langle P^* - P_t, C \rangle \leq \frac{\delta(C)}{4}$ is a sufficient condition for $P_{t+1}$ to be rank $k$. We are going to show that this sufficient condition holds for all $t$ by using Theorem 4.1. In particular it is sufficient to have that $\frac{k^2 \log(1/\delta)(\log(T))^2}{\sqrt{T + \frac{1}{\epsilon}}} \leq O(\frac{\delta(C)}{k})$, or equivalently $\sqrt{\epsilon} \leq O\left( \frac{\delta(C)}{k^3 \log(1/\delta)} \right)$. This implies that the conditions of Theorem 4.1 are met whenever $\|P_1 - P^*\|_F^2 \leq \frac{4k\|C - C_0\|^2}{\delta(C)^2} \leq \epsilon = O\left( \left( \frac{\delta(C)}{k^3 \log(1/\delta)} \right)^2 \right)$ and $\|C_t - C\| \leq \frac{\sqrt{\epsilon}}{\log(1/\epsilon)} = O\left( \frac{\delta(C)}{k^3 \log(1/\delta)} \frac{1}{\log\left( \frac{\delta(C)}{k^3 \log(1/\delta)} \right)} \right)$. Matrix Bernstein implies that the first inequality is satisfied with probability $1 - \delta$, whenever we use $n_1 = \Theta\left( \frac{\log(d/\delta)k^2 \log(1/\delta)}{\delta(C)^3} \right)$. Since we require that the second inequality hold for all $t \leq T$, Matrix Bernstein together with a union bound guarantees that the inequality holds as long as we use $n_2 = \Theta\left( \frac{\log(T) \log(d/\delta)k^3 \log(1/\delta) \log(\delta(C)/(k^3 \log(1/\delta)))}{\delta(C)^2} \right)$. $\square$

# B  Proofs for MB-RMSG

## B.1  Sufficient condition

We now consider the update $P_{t+1/2} = (1 - \lambda\eta_t)P_t + \eta_t C_t$ of MB-RMSG in Algorithm 2.

**Lemma B.1.** *Let $P_t$ be rank $k$ and suppose $\|C - C_t\| \leq \epsilon$. Then a sufficient condition for $P_{t+1}$ to be rank $k$ is*

$$\langle P_t, C \rangle \geq \langle P^*, C \rangle - \frac{\delta(C)}{2} + \frac{\lambda}{2} + \epsilon(k+1). \tag{12}$$

*Proof of Lemma B.1.* Let $\alpha_t = 1 - \lambda\eta_t$ and write $P_{t+1/2} = U_{t+1/2} \begin{pmatrix} \alpha_t + \lambda_1 & & & & & \\ & \alpha_t + \lambda_2 & & & & \\ & & \ddots & & & \\ & & & \lambda_{k+1} & & \\ & & & & \ddots & \\ & & & & & \lambda_d \end{pmatrix} U_{t+1/2}^\top$. Let $y = \begin{pmatrix} \alpha_t + \lambda_1 \\ \alpha_t + \lambda_2 \\ \vdots \\ \lambda_d \end{pmatrix}$. As before, the shift if we would like to retain rank $k$ of the iterates is given by $s_{i^*,j^*} = \frac{\sum_{l=i^*+1}^{k} y_l - k + i^*}{k - i^* - 1}$ and thus a sufficient condition is $\frac{\sum_{l=i^*+1}^{k} y_l - k + i^*}{k - i^* - 1} \geq \lambda_{k+1}$. Slightly rewriting the sum in the numerator we get

$$\frac{\sum_{l=i^*+1}^{k} y_l - k + i^*}{k - i^* - 1} = \frac{\sum_{l=i^*+1}^{k} \lambda_l + (k - i^*)\alpha_t - (k - i)^*}{k - i^* - 1} = \frac{\sum_{l=i^*+1}^{k} \lambda_l - (k - i^*)(1 - \alpha_t)}{k - i^* - 1}.$$

Thus a sufficient condition becomes $\frac{\sum_{l=i^*+1}^{k} \lambda_l}{k - i^*} \geq \lambda_{k+1} + (1 - \alpha_t)$. Using the average is larger than the minimum we convert this sufficient condition to $\lambda_k \geq \lambda_{k+1} + (1 - \alpha_t)$ or written equivalently

$\lambda_k(\mathrm{P}_{t+1/2}) \geq \lambda_{k+1} + (1-\alpha_t) + \alpha_t = 1 + \lambda_{k+1}$. Now we write

$$\lambda_k(\mathrm{P}_{t+1/2}) = \max_{\mathrm{U} \in SO(k)} \lambda_{min}(\mathrm{U}^\top \mathrm{P}_{t+1/2}\mathrm{U}) \geq \lambda_{min}(\mathrm{U}_t^\top \mathrm{P}_{t+1/2}\mathrm{U}_t) = \lambda_k(\alpha_t \mathrm{I} + \eta_t \mathrm{U}_t^\top \mathrm{C}_t \mathrm{U}_t)$$

$$= \alpha_t + \eta_t \lambda_k(\mathrm{U}_t^\top \mathrm{C}_t \mathrm{U}_t).$$

Thus a sufficient condition becomes $\eta_t \lambda_k(\mathrm{U}_t^\top \mathrm{C}_t \mathrm{U}_t) \geq \lambda_{k+1} + 1 - \alpha_t = \lambda_{k+1} + \lambda\eta_t$. We now upper bound $\lambda_{k+1} = \lambda_{k+1}(\mathrm{P}_{t+1/2})$.

$$\lambda_{k+1} \leq \sum_{l=1}^{k+1} \lambda_l(\mathrm{P}_{t+1/2}) - \sum_{l=1}^{k} \lambda_l(\mathrm{U}_t^\top \mathrm{P}_{t+1/2}\mathrm{U}_t) \leq \alpha_t k + \eta_t \sum_{l=1}^{k+1} \lambda_l(\mathrm{C}_t) - \alpha_t k - \eta_t \sum_{l=1}^{k} \lambda_l(\mathrm{U}_t^\top \mathrm{C}_t \mathrm{U}_t)$$

$$= \eta_t \left\{ \sum_{l=1}^{k+1} \lambda_l(\mathrm{C}_t) - \sum_{l=1}^{k} \lambda_l(\mathrm{U}_t^\top \mathrm{C}_t \mathrm{U}_t) \right\}.$$

This implies that a sufficient condition is $\lambda_k(\mathrm{U}_t^\top \mathrm{C}_t \mathrm{U}_t) + \sum_{l=1}^{k} \lambda_l(\mathrm{U}_t^\top \mathrm{C}_t \mathrm{U}_t) \geq \sum_{l=1}^{k+1} \lambda_l(\mathrm{C}_t) + \lambda$. Assume that $\|\mathrm{C} - \mathrm{C}_t\| \leq \epsilon$ and repeat the proof of Lemma 5.2 to get that a sufficient condition is $\lambda_k(\mathrm{U}_t^\top \mathrm{C}\mathrm{U}_t) + \sum_{l=1}^{k} \lambda_l(\mathrm{U}_t^\top \mathrm{C}\mathrm{U}_t) - 2\epsilon(k+1) \geq \sum_{l=1}^{k+1} \lambda_l(\mathrm{C}) + \lambda$.

$$\lambda_k(\mathrm{U}_t^\top \mathrm{C}\mathrm{U}_t) + \sum_{l=1}^{k} \lambda_l(\mathrm{U}_t^\top \mathrm{C}\mathrm{U}_t) - 2\epsilon(k+1) \geq \sum_{l=1}^{k+1} \lambda_l(\mathrm{C}) + \lambda$$

$$\Longleftrightarrow$$

$$2\sum_{l=1}^{k} \lambda_l(\mathrm{U}_t^\top \mathrm{C}\mathrm{U}_t) - 2\epsilon(k+1) \geq \sum_{l=1}^{k-1} \lambda_l(\mathrm{C}) + \lambda_l(\mathrm{U}_t^\top \mathrm{C}\mathrm{U}_t) + \lambda_k(\mathrm{C}) + \lambda_{k+1}(\mathrm{C}) + \lambda$$

$$\Longleftarrow$$

$$2\sum_{l=1}^{k} \lambda_l(\mathrm{U}_t^\top \mathrm{C}\mathrm{U}_t) - 2\epsilon(k+1) \geq 2\sum_{l=1}^{k} \lambda_l(\mathrm{C}) - (\lambda_k(\mathrm{C}) - \lambda_{k+1}(\mathrm{C})) + \lambda$$

$$\Longleftrightarrow$$

$$\sum_{l=1}^{k} \lambda_l(\mathrm{U}_t^\top \mathrm{C}\mathrm{U}_t) - \epsilon(k+1) \geq \sum_{l=1}^{k} \lambda_l(\mathrm{C}) - \frac{\delta(\mathrm{C})}{2} + \frac{\lambda}{2}$$

$$\Longleftrightarrow$$

$$\langle \mathrm{P}_t, \mathrm{C} \rangle \geq \langle \mathrm{P}^*, \mathrm{C} \rangle - \frac{\delta(\mathrm{C})}{2} + \frac{\lambda}{2} + \epsilon(k+1)$$

$\square$

From Mianjy & Arora (2018) we know that $\lambda < \delta(C)$ and so we can choose $\lambda = \delta(C)/2$ to get a sufficient condition of the form

$$\langle \mathrm{P}_t, \mathrm{C} \rangle \geq \langle \mathrm{P}^*, \mathrm{C} \rangle - \frac{\delta(\mathrm{C})}{4} + \epsilon(k+1).$$

## B.2 Convergence of SGD from "warm start" for strongly convex functions

We are now going to extend Theorem 7.5 from Harvey et al. (2018) to the case when $\|\mathrm{x}_1 - \mathrm{x}^*\|^2 \leq \epsilon$ and we run SGD with step size $\eta_t = \frac{1}{\lambda(1+\frac{1}{\epsilon})}$, for $\lambda$-strongly convex and 1-Lipschitz function $f$. We begin by restating Theorem 4.1 from Harvey et al. (2018).

**Theorem B.2.** *Let $\alpha_t \in [0,1)$ and $\beta_t, \gamma_t \geq 0$ for all $t$ and let $K = \max_t \left( \frac{2\gamma_t}{1-\alpha_t}, \frac{2\beta_t^2}{1-\alpha_t} \right)$. Let $(X_t)_{t=1}^T$ be a stochastic process and let $(\mathcal{F}_t)_{t=1}^T$ be a filtration such that $X_t$ is $\mathcal{F}_t$ measurable and $X_t$ is non-negative a.s. Let $\widehat{w}_t$ be a mean-zero random variable conditioned on $\mathcal{F}_t$ such that $|\widehat{w}_t| \leq 1$ a.s. for all $t$. Suppose that $X_1 \leq K$ and that $X_{t+1} \leq \alpha_t X_t + \beta_t \widehat{w}_t \sqrt{X_t} + \gamma_t$ for all $t$. Then*

$$\mathbb{P}\left[ X_t \geq K \log(1/\delta) \right] \leq e\delta$$

*for all $t$.*

Assume that $\|\nabla f(x_t) - g_t\| \leq 1$ and $\|g_t\| \leq 1$. With the help of this theorem we can show the following.

**Lemma B.3.** *Suppose* $\|x_1 - x^*\|^2 \leq \epsilon$ *and we run SGD with step size* $\eta_t = \frac{1}{\lambda(t + \frac{1}{\epsilon})}$. *Then for all* $t$

$$\mathbb{P}\left[\|x_t - x^*\|^2 \geq \frac{8}{\lambda^2(t + \frac{1}{\epsilon} - 1)}\log(1/\delta)\right] \leq e\delta$$

*Proof of Lemma B.3.* Let $z_t = \nabla f(x_t) - g_t$. Repeating the proof of Lemma 6 in Rakhlin et al. (2012) we have

$$
\begin{aligned}
\|x_{t+1} - x^*\|^2 &\leq \|x_t - \eta_t g_t - x^*\|^2 = \|x_t - x^*\|^2 - 2\eta_t\langle g_t, x_t - x^*\rangle + \eta_t^2\|g_t\|^2 \\
&= \|x_t - x^*\|^2 - 2\eta_t\langle\nabla f(x)_t, x_t - x^*\rangle + 2\eta_t\langle z_t, x_t - x^*\rangle + \eta_t^2\|g_t\|^2 \\
&\leq \|x_t - x^*\|^2 - 2\eta_t(f(x_t) - f(x^*)) + \frac{\eta_t\lambda}{2}\|x_t - x^*\|^2 + 2\eta_t\langle z_t, x_t - x^*\rangle + \eta_t^2\|g_t\|^2 \\
&\leq \|x_t - x^*\|^2 - 2\eta_t\frac{\lambda}{2}\|x_t - x^*\|^2 + \frac{\eta_t\lambda}{2}\|x_t - x^*\|^2 + 2\eta_t\langle z_t, x_t - x^*\rangle + \eta_t^2\|g_t\|^2 \\
&\leq (1 - 2\eta_t\lambda)\|x_t - x^*\|^2 + 2\eta_t\langle z_t, x_t - x^*\rangle + \eta_t^2 \\
&= \left(1 - \frac{2}{t + \frac{1}{\epsilon}}\right)\|x_t - x^*\|^2 + \frac{2}{\lambda(t + \frac{1}{\epsilon})}\langle z_t, x_t - x^*\rangle + \frac{1}{\lambda^2(t + \frac{1}{\epsilon})^2}.
\end{aligned}
$$

Let $X_t = (t + \frac{1}{\epsilon} - 1)\|x_t - x^*\|^2$, then from the above we have

$$X_{t+1} \leq \frac{t + \frac{1}{\epsilon} - 2}{t + \frac{1}{\epsilon} - 1}X_t + \frac{2}{\lambda}\frac{\langle z_t, x_t - x^*\rangle}{\|x_t - x^*\|}\sqrt{\frac{X_t}{t + \frac{1}{\epsilon} - 1}} + \frac{1}{\lambda^2(t + \frac{1}{\epsilon})}.$$

Let $\alpha_t = \frac{t + \frac{1}{\epsilon} - 2}{t + \frac{1}{\epsilon} - 1}$, $\widehat{w}_t = \frac{\langle z_t, x_t - x^*\rangle}{\|x_t - x^*\|}$, $\beta_t = \frac{2}{\lambda\sqrt{t + \frac{1}{\epsilon} - 1}}$ and $\gamma_t = \frac{1}{\lambda^2(t + \frac{1}{\epsilon})}$. Then $K \leq \frac{8}{\lambda^2}$ and $X_1 = \frac{1}{\epsilon}\|x_t - x^*\|^2 \leq 1 < \frac{8}{\lambda^2}$. Applying Theorem B.2 finishes the proof. $\square$

**Theorem B.4** (Theorem 2 Yu et al. (2014)). *Let* $C, \widehat{C} \in \mathbb{R}^{d \times d}$ *be symmetric, with eigenvalues* $\lambda_1 \geq \ldots \geq \lambda_d$ *and* $\widehat{\lambda}_1, \ldots, \widehat{\lambda}_d$. *Fix* $j \in \{1, \ldots, d\}$, *and assume that* $\min(\lambda_{j-1} - \lambda_j, \lambda_j - \lambda_{j+1}) > 0$, *where* $\lambda_0 = \infty, \lambda_{d+1} = -\infty$. *If* $V, \widehat{V}$ *are the orthogonal matrices corresponding to the top* $k$ *eigenvalues of* $C$ *and* $\widehat{C}$ *respectively, then*

$$\sqrt{k - \|V^\top\widehat{V}\|_F^2} \leq \frac{2\sqrt{k}\|\widehat{C} - C\|}{\delta(C)},$$

*where* $\delta(C) = \lambda_k(C) - \lambda_{k+1}(C)$.

**Theorem B.5.** *Assume* $\|x_{t_n}\|^2 \leq \frac{1}{2}$. *After running Algorithm 2, with probability at least* $1 - 3e\delta$, *it holds that*

$$\langle P^* - P_T, C\rangle \leq \frac{32\log(1/\delta)}{\delta(C)^2(T + \frac{1}{\epsilon} - 1)}$$

*and for all* $t \leq T$ *it holds that* $P_t$ *is a rank-$k$ projection matrix, where* $\epsilon = \frac{\delta(C)^3}{128\log(1/\delta)}$.

*Proof of Theorem 4.2.* For all $1 \leq t \leq T - 1$, as long as $\|C - C_t\| \leq \frac{\delta(C)}{8(k+1)}$, Lemma B.1 implies that a sufficient condition for $P_{t+1}$ to be a rank-$k$ projection matrix, given $P_t$ is a rank-$k$ projection matrix, is $\langle P^* - P_t, C\rangle \leq \frac{\delta(C)}{8}$. Using Matrix Bernstein's inequality we know that for each $t$ we have $\mathbb{P}\left[\|C - C_t\| \geq \frac{\delta(C)}{8(k+1)}\right] \leq \frac{\delta}{T}$. On the other hand we know from the proof of Theorem 2.5 in Mianjy & Arora (2018) that $\langle P^* - P_t, C\rangle \leq \frac{\lambda_1(C)}{2}\|P_t - P^*\|_F^2 \leq \frac{\|P_t - P^*\|_F^2}{2}$. Lemma B.3 implies that

$$\mathbb{P}\left[\langle P^* - P_t, C\rangle \geq \frac{4}{\lambda^2(t + \frac{1}{\epsilon} - 1)}\log(1/\delta)\right] \leq e\delta,$$

as long as $\|P_1 - P^*\|_F^2 \leq \tilde{\epsilon}$, where $\lambda = \frac{\delta(C)}{2}$. Thus we only need to guarantee that $\frac{4\log(1/\delta)\tilde{\epsilon}}{\lambda^2} \leq \frac{\lambda}{2}$ or equivalently $\tilde{\epsilon} \leq \frac{\delta(C)^3}{128\log(1/\delta)}$. To bound $\|P_1 - P^*\|_F^2$ we use Theorem B.4. Let $P_1 = UU^\top$ and $P^* = U^*(U^*)^\top$, then

$$\|P_1 - P^*\|_F^2 = 2(k - \langle P_1, P^* \rangle) = 2(k - \mathrm{Tr}\left(UU^\top U^*(U^*)^\top\right))$$

$$= 2(k - \mathrm{Tr}\left((U^*)^\top UU^\top U^*\right) = 2(k - \|U^\top U_*\|_F^2) \leq 4k\frac{\|C - C_0\|^2}{\delta(C)^2}.$$

And so we require $4k\frac{\|C-C_0\|^2}{\delta(C)^2} \leq \frac{\delta(C)^3}{128\log(1/\delta)}$ or equivalently $\|C - C_0\| \leq \frac{\delta(C)^{5/2}}{16\sqrt{2k\log(1/\delta)}}$. Using Matrix Bernstein's inequality this is guaranteed to hold with probability at least $1 - \delta$ as long as $n \geq \log(d/\delta)\frac{128k\log(1/\delta)}{\delta(C)^5}$. A union bound now guarantees that for every $t \leq T$, $P_t$ is a rank-$k$ matrix and together with the projection onto the rescaled simplex, this guarantees that $P_t$ is a rank-$k$ projection matrix. $\square$

### B.3 Improved rates in expectation

For this subsection, the expectation we take is the conditional expectation on the event that $P_t$ is a rank-$k$ projection matrix and that $\|C_t - C\| \leq \frac{\delta(C)}{8(k+1)}$ for all $t \in [T]$. We are now going to take a proximal view of projected SGD of Algorithm 2. In particular we let $f(P) = -\langle P, C \rangle + \frac{\lambda}{4}\|P\|_F^2$ and $\psi(P) = \frac{\lambda}{4}\|P\|_F^2 + \chi_{\mathcal{P}}(P)$, where $\chi_{\mathcal{P}}(P)$ is the characteristic function of the set of constraints $\mathcal{P} = \{P : \mathrm{Tr}(P) \leq k, 0 \preceq P \preceq I, P^\top = P\}$. Notice that $f$ is a convex and $\frac{2}{\lambda}$-smooth with respect to Frobenius norm and that $\psi$ is $\frac{\lambda}{2}$-strongly convex with respect to Frobenius norm. We can write the update of Algorithm 2 as

$$P_{t+1} = \arg\min_{P \in \mathbb{R}^{d \times d}} \frac{1}{2\eta_t}\|P - P_t\|_F^2 + \langle g_t, P - P_t \rangle + \psi(P) - \psi(P_t),$$

where $g_t$ is the stochastic gradient of $f$ at $P_t$ based on $C_t$. Let

$$Prog(P_t) = -\min_{P \in \mathbb{R}^{d \times d}} \frac{1}{2\eta_t}\|P - P_t\|_F^2 + \langle g_t, P - P_t \rangle + \psi(P) - \psi(P_t).$$

We have

$$Prog(P_t) = -\left\{\frac{1}{2\eta_t}\|P_{t+1} - P_t\|_F^2 + \langle g_t, P_{t+1} - P_t \rangle + \psi(P_{t+1}) - \psi(P_t)\right\}$$

$$= -\left\{\frac{1}{2\eta_t}\|P_{t+1} - P_t\|_F^2 + \langle \nabla f(P_t), P_{t+1} - P_t \rangle + \psi(P_{t+1}) - \psi(P_t) + \langle g_t - \nabla f(P_t), P_{t+1} - P_t \rangle\right\}$$

$$\leq -\left\{\frac{1}{2\eta_t}\|P_{t+1} - P_t\|_F^2 - \frac{L}{2}\|P_{t+1} - P_t\|_F^2 + f(P_{t+1}) - f(P_t)\right.$$

$$\left. + \psi(P_{t+1}) - \psi(P_t) + \langle g_t - \nabla f(P_t), P_{t+1} - P_t \rangle\right\},$$

where $L$ is the smoothness parameter of $f$ and the inequality follows from smoothness. Let $\beta_t = \frac{1}{\eta_t} - \frac{L}{2}$ and notice that

$$\langle \nabla f(P_t) - g_t, P_{t+1} - P_t \rangle - \frac{\beta_t}{2}\|P_{t+1} - P_t\|_F^2 \leq \frac{1}{2\beta_t}\|\nabla f(P_t) - g_t\|_F^2.$$

This implies that

$$f(P_t) - f(P_{t+1}) + \psi(P_t) - \psi(P_{t+1}) \geq Prog(P_t) - \frac{1}{2\beta_t}\|\nabla f(P_t) - g_t\|_F^2.$$

Lemma 2.5 from Allen-Zhu (2017) implies that

$$\eta_t\langle g_t, P_{t+1} - P \rangle + \eta_t\psi(P_{t+1}) - \eta_t\psi(P) \leq -\frac{1}{2}\|P_t - P_{t+1}\|_F^2 + \frac{1}{2}\|P_t - P\|_F^2 - \frac{1 + \eta_t\sigma}{2}\|P_{t+1} - P\|_F^2,$$

where $\sigma$ is the strong convexity parameter of $\psi$ for any rank-$k$ projection matrix P. By construction we know that $f$ is also $\sigma$ strongly convex and thus we have

$$\mathbb{E}\left[\langle g_t, P_t - P \rangle\right] \geq \mathbb{E}\left[f(P_t) - f(P) + \frac{\sigma}{2}\|P_t - P\|_F^2\right].$$

Combining the above two inequalities we arrive at

$$\mathbb{E}\left[\eta_t\left(f(\mathrm{P}_t) - f(\mathrm{P}) + \frac{\sigma}{2}\|\mathrm{P}_t - \mathrm{P}\|_F^2\right)\right] + \eta_t\left(\langle g_t, \mathrm{P}_{t+1} - \mathrm{P}_t\rangle + \psi(\mathrm{P}_{t+1}) - \psi(\mathrm{P})\right)$$
$$\leq -\frac{1}{2}\|\mathrm{P}_t - \mathrm{P}_{t+1}\|_F^2 + \frac{1}{2}\|\mathrm{P}_t - \mathrm{P}\|_F^2 - \frac{1 + \eta_t\sigma}{2}\|\mathrm{P}_{t+1} - \mathrm{P}\|_F^2.$$

Next we use the property that $\mathrm{P}_{t+1}$ is a rank-$k$ projection to write

$$\eta_t\left(\langle g_t, \mathrm{P}_{t+1} - \mathrm{P}_t\rangle + \psi(\mathrm{P}_{t+1}) - \psi(\mathrm{P})\right) = \eta_t\langle g_t, \mathrm{P}_{t+1} - \mathrm{P}_t\rangle + \psi(\mathrm{P}_{t+1}) - \psi(\mathrm{P}),$$

which implies

$$\mathbb{E}\left[\eta_t\left(f(\mathrm{P}_t) - f(\mathrm{P}) + \frac{\sigma}{2}\|\mathrm{P}_t - \mathrm{P}\|_F^2\right)\right] \leq -\left\{\frac{1}{2}\|\mathrm{P}_t - \mathrm{P}_{t+1}\|_F^2 + \eta_t\langle g_t, \mathrm{P}_{t+1} - \mathrm{P}_t\rangle + \psi(\mathrm{P}_{t+1}) - \psi(\mathrm{P})\right\}$$
$$+ \frac{1}{2}\|\mathrm{P}_t - \mathrm{P}\|_F^2 - \frac{1 + \eta_t\sigma}{2}\|\mathrm{P}_{t+1} - \mathrm{P}\|_F^2.$$

Again using the fact that $\psi(P_t) = \psi(\mathrm{P})$ we have

$$\mathbb{E}\left[\eta_t(f(\mathrm{P}_t) - f(\mathrm{P}))\right] \leq \mathbb{E}\left[\eta_t Prog(\mathrm{P}_t) + \frac{1 - \eta_t\sigma}{2}\|\mathrm{P}_t - \mathrm{P}\|_F^2 - \frac{1 + \eta_t\sigma}{2}\|\mathrm{P}_{t+1} - \mathrm{P}\|_F^2\right].$$

Using the $Prog$ inequality from earlier we have

$$\mathbb{E}\left[\eta_t(f(\mathrm{P}_t) - f(\mathrm{P}))\right] \leq \mathbb{E}\left[\eta_t(f(\mathrm{P}_t) - f(\mathrm{P}_{t+1})) + \frac{1 - \eta_t\sigma}{2}\|\mathrm{P}_t - \mathrm{P}\|_F^2 - \frac{1 + \eta_t\sigma}{2}\|\mathrm{P}_{t+1} - \mathrm{P}\|_F^2\right] + \frac{\eta_t}{2\beta_t}V_t^2,$$

where $V_t = \mathbb{E}\left[\|\nabla f(\mathrm{P}_t) - g_t\|_F\right]$. The above implies

$$\mathbb{E}\left[f(\mathrm{P}_{t+1}) - f(\mathrm{P})\right] \leq \mathbb{E}\left[\frac{1 - \eta_t\sigma}{2\eta_t}\|\mathrm{P}_t - \mathrm{P}\|_F^2 - \frac{1 + \eta_t\sigma}{2\eta_t}\|\mathrm{P}_{t+1} - \mathrm{P}\|_F^2 + \frac{1}{2\beta_t}V_t^2.\right]$$

Recall that we set the step size as $\eta_t = \frac{2}{\sigma(t + 1/\epsilon)}$. This implies

$$\frac{1 - \eta_t\sigma}{2\eta_t}\|\mathrm{P}_t - \mathrm{P}\|_F^2 - \frac{1 + \eta_{t-1}\sigma}{2\eta_{t-1}}\|\mathrm{P}_t - \mathrm{P}\|_F^2 = \frac{1}{2}\|\mathrm{P}_t - \mathrm{P}\|_F^2\left(\frac{\sigma(t + 1/\epsilon)}{2} - \frac{\sigma(t - 1 + 1/\epsilon)}{2} - 2\sigma\right)$$
$$= -\frac{3}{4}\sigma\|\mathrm{P}_t - \mathrm{P}\|_F^2 < 0.$$

Recall that $\sigma = \frac{\delta(\mathrm{C})}{4}$ and from our initial condition we know that $f(\mathrm{P}_1) - f(\mathrm{P}^*) \leq \frac{\delta(\mathrm{C})}{8}$. Thus summing from $t = 2$ to $T - 1$ we have

$$\mathbb{E}\left[\frac{1}{T-1}\sum_{t=1}^{T} f(\mathrm{P}_t) - f(\mathrm{P}^*)\right] \leq \frac{\sigma}{T-1} + \frac{1}{T-1}\sum_{t=2}^{T-1}\frac{V_t^2}{2\beta_t}.$$

Let us upper bound $\frac{1}{\beta_t}$.

$$\frac{1}{\beta_t} = \frac{1}{1/\eta_t - L} = \frac{1}{\frac{\sigma(t + 1/\epsilon)}{2} - \frac{2}{\sigma}} = \frac{2}{\sigma t + \frac{\sigma}{\epsilon} - \frac{4}{\sigma}} \leq \frac{2}{\sigma t}.$$

Where the last inequality follows since $\sigma = \frac{\delta(\mathrm{C})}{4}$ and $\epsilon < \frac{\delta(\mathrm{C})^3}{128}$. Next we upper bound $V_t^2$. Recall that we conditioned on the event that $\|\mathrm{C}_t - \mathrm{C}\| \leq \frac{\delta(\mathrm{C})}{8(k+1)}$ for all $t \in [T]$. We also assumed that $\mathrm{Tr}(\mathrm{C}_t) \leq 1$. Thus

$$\|\nabla f(\mathrm{P}_t) - g_t\|_F^2 = \|\mathrm{C}_t - \mathrm{C}\|_F^2 \leq 2\mathrm{Tr}(\mathrm{C})\|\mathrm{C}_t - \mathrm{C}\| \leq \frac{\sigma}{2k}.$$

This implies we can bound $\sum_{t=2}^{T-1}\frac{V_t^2}{2\beta_t} \leq \frac{\log(T)}{k}$. It is interesting to note that we could also bound the above sum by $\frac{d\sigma\log(T)}{k}$ and get a convergence rate depending on $d\sigma$ instead. We are now going to use the trick in Shamir & Zhang (2013) to obtain a convergence guarantee for the last iterate. Instead

of replacing P by $P^*$ and summing from $t = 2$ to $T - 1$. we sum from $t = T - l + 1$ to $T$ and set $P = P_{T-l}$. Thus we get

$$\mathbb{E}\left[\sum_{t=T-l+1}^{T} f(P_{t+1}) - f(P_{T-l})\right] \leq \sum_{t=T-l}^{T-1} \frac{V_t^2}{2\beta_t}.$$

Set $S_l = \frac{1}{l}\sum_{t=T-l+1}^{T} f(P_t)$. We have

$$-\mathbb{E}\left[f(P_{T-l})\right] \leq -\mathbb{E}\left[S_l\right] + \frac{1}{l}\sum_{t=T-l}^{T-1} \frac{V_t^2}{2\beta_t}.$$

Using the definition of $S_l$ we also have

$$(l-1)S_{l-1} = lS_l - f(P_{T-l+1}) \leq lS_l - S_{l-1} + \frac{1}{l-1}\sum_{t=T-l+1}^{T-1} \frac{V_t^2}{2\beta_t}$$

$$\iff$$

$$lS_{l-1} \leq lS_l + \frac{1}{l-1}\sum_{t=T-l+1}^{T-1} \frac{V_t^2}{2\beta_t}$$

$$\iff$$

$$S_{l-1} \leq S_l + \frac{1}{l(l-1)}\sum_{t=T-l+1}^{T-1} \frac{V_t^2}{2\beta_t}.$$

Thus $\mathbb{E}\left[f(P_T)\right] \leq \mathbb{E}\left[S_T\right] + \sum_{l=2}^{T} \frac{l}{l(l-1)}\sum_{t=T-l+1}^{T-1} \frac{V_t^2}{2\beta_t} \leq \frac{\sigma}{T-1} + \frac{\log(T)}{(T-1)k} + \frac{(1+\log(T))}{(T-1)k}$. And so we have shown the following theorem.

**Theorem B.6.** *Let $\mathcal{A}$ be the event that for all $t \in [T]$ it holds that $\|C_t - C\| \leq \frac{\delta(C)}{8(k+1)}$ and $P_t$ is a rank-$k$ projection matrix. Then Algorithm 2 guarantees that $\mathcal{A}$ occurs with probability at least $1 - e\delta$ and that*

$$\mathbb{E}\left[\langle P^* - P_T, C\rangle | \mathcal{A}\right] \leq \tilde{O}\left(\frac{\delta(C)}{T} + \min(\delta(C) \times d, 1)\frac{1}{kT}\right).$$