[Reviews · NeurIPS 2019]

Reviewer 1



The authors propose mini-batched versions of matrix stochastic gradient (MSG) and regularized matric stochastic gradient (RMSG) for PCA. They show for recovering top $k$ the proposed methods match the per iteration complexity of the Oja's algorithm up to order k. They show rank control of the iterates in the proposed algorithms by deriving a sufficient condition on the rank of the iterates being $k$ in Lemma 5.2. Finally, some synthetic results are presented to verify the convergence and rank control for the proposed algorithm. Originality: The originality lies in introducing the mini-batching on top of MSG and R-MSG. Also, the paper introduces a novel sufficiency condition for rank control of the iterates of MB-MSG and MB-RMSG (Lemma 5.2). The previous rank control idea for the top 1 PCA in Garber et al. (2018) is claimed to be not easily extendable to top k PCA. Quality: The authors honestly present which ideas are existing and which ideas are new. The proofs are hard to follow at times as detailed in the Improvements part. I believe the proofs, however, I am not completely convinced of their correctness. Clarity: The exposition in the paper is clear. Significance: The paper takes an important step towards understanding the empirical performance of MSG and RMSG for the PCA problem. The results presented in the paper, work with mini-batched versions of MSG and RMSG. However, it is unclear how the results here can be extended for the general MSG and RMSG.

Reviewer 2



The work studies two algorithms (MG-MSG and MB-RMSG) on a convex relaxation to principal component analysis and gives new performance bounds. The analysis and experiments are solid. I recommend it to be accepted.

Reviewer 3



This work considers convex relaxation to the PCA problem and revisit two algorithms: matrix stochastic gradient (MSG) and l2 regularized MSG (RMSG) from the theoretical perspective. For the Algorithm 1, it is not clear on how to choose T, the number of iterations. For Theorem 4.1, it is a bit confusing that the upper bound will get large as the value of T increases. Note that T is the number of iterations. One would expect that a larger T should lead to tight bound. For the numerical study, it is better to compare some existing methods to have fair evaluation of the proposed method.

[Author Response · NeurIPS 2019]

**Reviewer 1:** *"The implementation details should be presented along with the algorithm as they seem to be necessary for claiming the stated per iteration complexity."* – The implementation details are in Section 6, if the reviewer believes that readability of the paper would be improved, we can include a sketch of the main implementation details when we introduce the algorithms.

*"Clarifications of the proof: The authors should explain how Ky Fan's inequality is used in the derivation of Eq. 8 in the supplementary material. What is eta in Eq. 8?"* – $\eta$ is a typo and should be $\eta_t$. Ky Fan's inequality states that $\sum_{l=1}^{k+1} \lambda_l(P_t + \eta_t C_t) \leq \sum_{l=1}^{k+1} \lambda_l(P_t) + \sum_{l=1}^{k+1} \eta_t \lambda_l(C_t)$. Since by assumption $P_t$ is a rank-k projection matrix $\sum_{l=1}^{k+1} \lambda_l(P_t) = k$ and this is how the inequality holds.

*"Some explanations about how Eq. 9 follows from 1) Eq. 8, 2) inequality of $\lambda_k(P_{t+1/2})$, and the relation between $\lambda_k(P_{t+1/2})$ and $\lambda_{k+1}(P_{t+1/2})$."* – From the discussion up to line 354 we have that $1 + \lambda_{k+1}(P_{t+1/2}) \leq \lambda_k(P_{t+1/2})$ is a sufficient condition. Now Eq. 8) implies that a sufficient condition is $1 + \eta_t \sum_{l=1}^{k+1} \lambda_l(C_t) - \eta_t \sum_{l=1}^{k+1} \lambda_l(U_t^\top C_t U_t) \leq \lambda_k(P_{t+1/2})$. Finally since it always holds that $\lambda_k(P_{t+1/2}) \geq 1 + \eta_t \lambda_k(U_t^\top C_t U_t)$ we get that a sufficient condition is $1 + \eta_t \sum_{l=1}^{k+1} \lambda_l(C_t) - \eta_t \sum_{l=1}^{k+1} \lambda_l(U_t^\top C_t U_t) \leq 1 + \eta_t \lambda_k(U_t^\top C_t U_t)$ or equivalently Eq. 9)

*"For the proof of Lemma A.2, the inequality in line 365-366 seems to be stricter than Lemma 5.1. How do we obtain that using Lemma 5.1? Similar concerns hold for the proof of Lemma B.1. It is important to clarify this step."* – First note that Eq. 9) and statement of Lemma 5.1 are equivalent. The inequality on lines 365-366 follows by replacing $\sum_{l=1}^{k}(U_t^\top C_t U_t) + \lambda_k(U_t^\top C_t U_t)$ in Eq. 9) by $\sum_{l=1}^{k}(U_t^\top C U_t) + \lambda_k(U_t^\top C U_t) + \epsilon(k+1)$, which can be done because of the derivation between lines 364 and 365. A similar derivation holds for Lemma B.1

We will add the above clarifications and other missing steps to the appendix.

**Reviewer 2:** We would like to thank the reviewer for the comments.

**Reviewer 3:** *"algorithmic contributions: Fair. Not very sure how computationally efficient of the developed algorithm."* – Algorithm 2 is as computationally efficient as Oja's algorithm up to a factor of $k$ (as Theorem 4.3 and the discussion after it state – lines 168-171) which is considered state of the art. We have further discussed settings in which Algorithm 2 can perform better than Oja.

*"For the Algorithm 1, it is not clear on how to choose T, the number of iterations."* – Theorem 4.1 suggests how T should be set. Indeed if we return the last iterate of the algorithm $P_T$, then the suboptimality in objective is going to be of order $\tilde{O}(1/\sqrt{T})$ (disregarding other terms). This implies that if we want to achieve $\epsilon$-suboptimality, we need to set $T \sim 1/\epsilon^2$.

*"For Theorem 4.1, it is a bit confusing that the upper bound will get large as the value of T increases. Note that T is the number of iterations. One would expect that a larger T should lead to tight bound."* – The upper bound grows with $T$ only if $t$ is fixed. Since practitioners usually use the last iterate of the algorithm (in this case $t = T$) the upper bound clearly decreases in this case as $O(\log(T)/\sqrt{T})$ (disregarding other terms).

*"More explanation on the theoretical results are needed. More comprehensive numerical study can be helpful to demonstrate the advantage of the proposed method."* – The main theorems are presented in standard form for the PCA problem i.e. they give a bound on the suboptimality in objective after running the respective algorithm for $T$ iterations. We already compare with 3 other state of the art methods on real as well as synthetic data. We can add experiments on a wider range of datasets in the final version of our work.

[Meta-Review · NeurIPS 2019]

The paper proposes a mini-batched versions of matrix stochastic gradient and regularized matric stochastic gradient for PCA. It presents two algorithms based on a convex relaxation to the PCA problem, with convergence guarantees for both of them. Numerical results could be improved.